# Circum-Antarctic abundance and properties of CCN and INP

Christian Tatzelt[1], Silvia Henning[1], André Welti[2], Andrea Baccarini[3,4], Markus Hartmann[1,5], Martin Gysel-Beer[4], Manuela van Pinxteren[1], Robin L. Modini[4], Julia Schmale[3,4], and Frank Stratmann[1]

[1]Leibniz Institute for Tropospheric Research, Permoserstrasse 15, 04318 Leipzig, Germany
[2]Finnish Meteorological Institute, Erik Palménin aukio 1, FI-00560 Helsinki, Finland
[3]Extreme Environments Research Laboratory, École Polytechnique Fédérale de Lausanne, School of Architecture, Civil and Environmental Engineering, Lausanne, Switzerland
[4]Laboratory of Atmospheric Chemistry, Paul Scherrer Institute, 5232 Villigen PSI, Switzerland
[5]Atmospheric Science, Department of Chemistry and Molecular Biology, University of Gothenburg, Gothenburg, Sweden

**Correspondence:** Silvia Henning (silvia.henning@tropos.de), Julia Schmale (julia.schmale@epfl.ch)

**Abstract.** Aerosol particles acting as cloud condensation nuclei (CCN) or ice nucleating particles (INP) play a major role in the formation and glaciation of clouds. Thereby they exert a strong impact on the radiation budget of the Earth. Data on abundance and properties of both types of particles are sparse, especially for remote areas of the world, such as the Southern Ocean (SO). In this work, we present unique results from ship-borne aerosol-particle-related in situ measurements and filter sampling in the SO region, carried out during the Antarctic Circumnavigation Expedition (ACE) in the Austral summer of 2016/17. An overview of CCN and INP concentrations on the Southern Ocean is provided and, using additional quantities, insights regarding possible CCN and INP sources and origins are presented. CCN number concentrations spanned 2 orders of magnitude, e.g., for a supersaturation of $0.3\,\%$ values ranged roughly from 3 to $590\,cm^{-3}$. CCN showed variable contributions of organic and inorganic material (inter-quartile range of hygroscopicity parameter $\kappa$ from 0.2 to 0.9). No distinct size-dependence of $\kappa$ was apparent, indicating homogeneous composition across sizes (critical dry diameter on average between 30 and $110\,nm$). The contribution of sea spray aerosol (SSA) to the CCN number concentration was on average small. Ambient INP number concentrations were measured in the temperature range from $-5$ to $-27°C$ using an immersion freezing method. Concentrations spanned up to 3 orders of magnitude, e.g., at $-16°C$ from 0.2 to $100\,m^{-3}$. Elevated values (above $10\,m^{-3}$ at $-16°C$) were measured when the research vessel was in the vicinity of land (excluding Antarctica), with lower and more constant concentrations when at sea. This, along with results of backward-trajectory analyses, hints towards terrestrial and/or coastal INP sources being dominant close to ice-free (non-Antarctic) land. In pristine marine areas INP may originate from both oceanic sources and/or long range transport. Sampled aerosol particles ($PM_{10}$) were analysed for sodium and methanesulfonic acid (MSA). Resulting mass concentrations were used as tracers for primary marine and secondary aerosol particles, respectively. Sodium, with an average concentration around $2.8\,\mu g\,m^{-3}$, was found to dominate the sampled particle mass. MSA was highly variable over the SO, with concentrations up to $0.5\,\mu g\,m^{-3}$ near the sea ice edge. A correlation analysis yielded strong correlations between sodium mass concentration and particle number concentration in the coarse mode, unsurprisingly indicating a significant contribution of SSA to that mode. CCN number concentration was highly correlated with the number concentration of Aitken and accumulation mode particles. This, together with a lack of correlation between sodium mass and Aitken and accumulation mode number concentrations, underlines the important contribution of non-SSA, probably secondarily formed particles, to the

CCN population. INP number concentrations did not significantly correlate with any other measured aerosol physico-chemical parameter.

## 1  Introduction

Earth's changing climate and the human influence on it are undeniable facts (IPCC, 2013). Emissions of greenhouse gases (e.g., carbon dioxide) and their impact on the radiation budget are well understood, with high confidence and low uncertainty.
A larger uncertainty emerges from the lack of knowledge on atmospheric aerosol particles, in particular their influence on cloud-radiative properties. As there are natural and anthropogenic aerosol sources, the human impact on aerosol-cloud interactions is difficult to quantify. One way of reducing the uncertainty concerning the human influence on atmospheric aerosol particles, pointed out by Carslaw et al. (2013), is better constraining conditions before human impact, in the preindustrial time. With an atmospheric general-circulation model, Hamilton et al. (2014) searched for still-existing regions with preindustrial-like
conditions, by comparing simulations of atmospheric conditions in 1750 and 2000. The Southern Ocean region was found to feature pristine aerosol conditions during the Southern hemisphere summer months, making it an excellent region for measurements of preindustrial-like aerosol conditions. This was one of the key motivations for the "Study of Preindustrial-like Aerosol Climate Effects" (ACE-SPACE; Schmale et al., 2019) project within the framework of the Antarctic Circumnavigation Expedition (ACE), which was conducted across all sectors of the SO in the Austral summer 2016/17.

The focus of this study is on aerosol particles that can modulate cloud micro-physical properties and hence affect the cloud albedo (Twomey, 1974) and lifetime (Albrecht, 1989). Aerosol particles can initiate cloud droplet formation at levels of supersaturation ($SS$) much lower than the supersaturation necessary for homogeneous droplet formation (Köhler, 1936). The $SS$ at which particles activate is dictated primarily by their size, but also their chemical composition (Dusek et al., 2006). Particles acting as nuclei for cloud droplet formation at atmospherically relevant (water vapour) supersaturation are commonly
referred to as cloud condensation nuclei (CCN). Another group of cloud property-altering aerosol particles are ice nucleating particles (INP), which initiate cloud droplet freezing above the point of homogeneous freezing ($-38°C$). In summary, CCN play an important role in the formation of clouds, while INP alter the phase state (frozen or liquid) of cloud droplets which affects cloud radiative properties (Vergara-Temprado et al., 2018). Furthermore, cloud glaciation influences the precipitation formation (Wegener, 1911) and dissipation of clouds (Albrecht, 1989). As a consequence, changes in cloud radiative properties
and cloud lifetime impact Earth's climate (Lindzen, 1990; Murray et al., 2012). With that, CCN and INP play an important role for both weather and climate, and respective observations are fundamental to estimate the progression of climate change.

    Of the few aerosol-related studies over the SO, the majority focused on physical aerosol particle properties and aerosol composition. During the first Aerosol Characterization Experiment (ACE-1) in the Australian sector of the SO in 1995, Quinn et al. (1998) found the marine boundary layer (MBL) aerosol population with a particle diameter ($D_p$) between $100$ and $300\,\mathrm{nm}$
(referred to as "accumulation mode") to be minimally influenced by sea salt and mainly comprised of non-sea salt (nss) sulfate, i.e. the fraction of total sulfate not associated with sea salt. The main sources of nss-sulfate are: 1. sulfur compounds derived from continental anthropogenic sources (Savoie and Prospero, 1989), 2. oxidation of atmospheric dimethyl sulphide (DMS)

(Covert et al., 1992; Raes, 1995), and 3. volcanic emissions. DMS is produced by marine microbial activity and emitted from the ocean into the atmosphere in the gas phase (Curran et al., 2003; Abram et al., 2010). Sulfate formation from DMS oxidation is a complex multi-step process involving several intermediate molecules. For the sake of brevity, we simplify the description of the processes with three main pathways: 1. sulfuric acid production from homogeneous gas phase oxidation of DMS followed by condensation, 2. sulfur dioxide production from homogeneous gas phase oxidation of DMS followed by heterogeneous oxidation of sulfur dioxide to sulfuric acid in the liquid phase, and 3. reactive uptake of DMS into aqueous solution or cloud droplets followed by heterogeneous oxidation (Chen et al., 2018).

Properties of Aitken ($D_\mathrm{p} = 10$–$100\,\mathrm{nm}$) and accumulation mode particles ($D_\mathrm{p} = 100$–$1000\,\mathrm{nm}$ in this study) in the SO region were found to be clearly dependent on air-mass origin, with two distinct air-masses (polar and maritime) being encountered during the British Southern Ocean (BSO) campaign (O'Dowd et al., 1997), and the Plankton-derived Emissions of trace Gases and Aerosols in the Southern Ocean (PEGASO) cruise (Dall'Osto et al., 2017; Fossum et al., 2018). The two air-masses featured distinctly different aerosol populations in terms of concentration and chemical composition.

Looking at the chemical composition in the larger particle size ranges, for ACE-1 the population of particles with $D_\mathrm{p} = 300$–$5000\,\mathrm{nm}$ (referred to as "coarse mode") was found to be dominated by sea salt, with sporadic and minor contributions from nss-sulfate. Variations in the coarse mode sea salt concentrations could only partially ($40\,\%$) be explained by local wind speeds (Quinn et al., 1998).

The concentrations of particles in the MBL of the SO that act as CCN were investigated by a smaller number of studies. Quinn et al. (2017) found a large portion of the Aitken mode to act as CCN at a $SS > 0.5\,\%$. Sea spray aerosol (SSA), a mix of sea salt particles and ocean-derived organic species (de Leeuw et al., 2011), was found to dominate the CCN population, but only at $SS = 0.1\,\%$ in the high latitudes (down to $70°\,\mathrm{S}$) of the Southern hemisphere (Quinn et al., 2017). Cases of polar air during PEGASO featured CCN number concentrations at $SS = 0.8\,\%$ ($N_\mathrm{CCN,0.8}$) of $217 \pm 31\,\mathrm{cm}^{-3}$, while maritime cases showed almost doubled concentrations ($420 \pm 168\,\mathrm{cm}^{-3}$).

It remains an open question how CCN abundance is distributed over the SO and what typical values are, especially during the pristine conditions of the Austral summer. Further, CCN properties and origin are of interest. It is know that new particle formation (NPF) in the free troposphere is an important source of CCN in the MBL and occurs frequently over the summertime SO (McCoy et al., 2021). To our knowledge, it is not known which process/source (e.g., NPF or SSA) governs the CCN population of the SO generally and what role horizontal and vertical atmospheric transport plays (Baccarini et al., 2021).

Studies of INP number concentration ($N_\mathrm{INP}$) and origin in the SO region started with immersion freezing experiments by Bigg (1973), who measured $N_\mathrm{INP}$ between $3$–$250\,\mathrm{m}^{-3}$ at $-15°\mathrm{C}$. Two recent cruises were conducted as part of the Cloud, Aerosols, Precipitation, Radiation and Atmospheric Composition campaign (CAPRICORN-I & II). For CAPRICORN-I, observed $N_\mathrm{INP}$ over the SO in the temperature range between $-12$ and $-31°\mathrm{C}$ varied between $0.04$ and $1000\,\mathrm{m}^{-3}$ (McCluskey et al., 2018a). For context, $N_\mathrm{INP}$ at $-20°\mathrm{C}$ was found to be lower by a factor of up to $100$ compared to Bigg (1973). Preliminary INP results for CAPRICORN-II are presented in McFarquhar et al. (2021) and underline the findings for CAPRICORN-I of low but highly variable $N_\mathrm{INP}$ values on the SO. They also investigated the contribution of biological INP using heat treatment methods, assuming biological INP to be heat-labile. McCluskey et al. (2018a) found INP on the SO to be mainly heat-resistant,

with contributions from heat-labile INP in the $-15$ to $-20°C$ temperature range. In Bigg (1973) it was hypothesised, based on the fact that INP concentrations did not increase significantly in the vicinity of Australia, that there was no influence of dust from the continent. Correlation of INP and ambient radon concentration was used to assess whether sampled INP had terrestrial or oceanic sources for the CAPRICORN-I cruise. The INP source potential of bubble bursting was characterised for CAPRICORN-I in McCluskey et al. (2018a), using seawater samples. They found that INP were from oceanic sources, aerosolized by bubble bursting. Additionally, Uetake et al. (2020) showed that bacteria sampled during CAPRICORN-II are mostly of marine origin, suggesting a restricted meridional transport of continental aerosol towards the SO. In consequence, a dominance of sea spray on the INP population in the SO's MBL was concluded.

However, data on INP abundance, spatial distribution, properties, and sources over the SO region remain sparse. Regayre et al. (2020) pointed out that already a small number of observations from the SO can effectively reduce model uncertainty more than hundreds of measurements in the Northern hemisphere, as current simulations are based on very few observations in the Southern hemisphere. This demonstrates a need for further field measurements of CCN and INP in the SO region.

Parts of the CCN and INP data set presented in this study have previously been presented in the overview on the ACE cruise in Schmale et al. (2019). Aerosol properties were found to be highly heterogeneous over the SO. The CCN abundance in the MBL showed a significant sea spray contribution in the strong westerly wind belt, while in the polynyas of the Ross and Amundsen Sea biogenic emissions are more important. INP abundance was shown to be lower on the SO than in Northern hemispheric marine air, with small differences between samples on the open ocean and close to the Antarctic coast. INP abundance was found to be similar to other studies on the SO (e.g., McCluskey et al., 2018a) but lower than historic data from Bigg (1973). INP concentrations at $-15°C$ from the ACE expedition have been presented in Welti et al. (2020). They show that $N_{\mathrm{INP}}$ from ship-based measurements are lowest in polar regions and highest in temperate climate zones. Overall, geographical variation in $N_{\mathrm{INP}}$ is below 2 orders of magnitude at any temperature. At low temperature, lower $N_{\mathrm{INP}}$ were encountered in the Southern hemisphere than in the Northern hemisphere and this was attributed to the concentration of dust particles active as INP. These two previous studies (Schmale et al., 2019; Welti et al., 2020) presented a subset of the CCN and the INP data in larger contexts. This paper focuses on the detailed analysis and interpretation of the observations, including $N_{\mathrm{CCN}}$ at all available $SS$ and $N_{\mathrm{INP}}$ at the full investigated temperature range. Based on 10-day backward-trajectories, an air-mass analysis was performed to locate potential INP sources. In addition, a correlation analysis was performed using CCN, INP, and additional data from the ACE expedition in order to find potential links between the measured properties.

## 2   Methods

Measurements were carried out in the framework of ACE (Walton and Thomas, 2018). The cruise took place between December 2016 and March 2017 on board the research vessel (RV) *Akademik Tryoshnikov*. Starting and ending in Cape Town (South Africa), the cruise was divided into three Legs: Cape Town to Hobart (Australia), Hobart to Punta Arenas (Chile) and Punta Arenas to Cape Town (Fig. 1). Several islands (Marion, Crozet, Kerguelen, Balleny, Scott, Peter 1$^{\mathrm{st}}$, Diego Ramierez, South

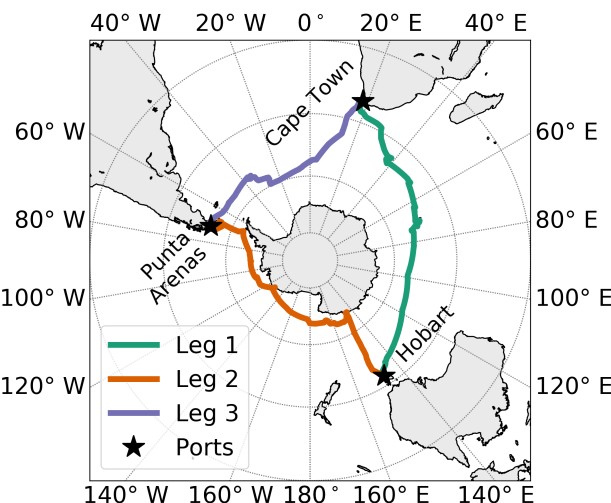

**Figure 1.** Hourly position of the RV *Akademik Tryoshnikov* during ACE. The ports visited as part of the cruise are indicated (stars). Leg 1 (green) from Cape Town (South Africa) to Hobart (Australia) between 20 December 2016 and 19 January 2017, Leg 2 (orange) from Hobart to Punta Arenas (Chile) between 22 January and 22 February 2017, and Leg 3 (purple) from Punta Arenas to Cape Town between 26 February and 19 March 2017.

Georgia, South Sandwich, and Bovetoya island), an Antarctic glacier (Mertz Glacier), and the Siple ice shelf were passed during ACE. At Mertz Glacier the southern-most latitude of $78°$ S was reached.

The instrumentation for real-time aerosol measurements was situated in a laboratory container on the fore-deck of the RV, equipped with two standard aerosol inlets (Global Atmosphere Watch; Weingartner et al., 1999) at roughly $15\,\mathrm{m}$ above sea level (a. s. l.), allowing for particles with $D_{\mathrm{p}} \leq 40\,\mathrm{\mu m}$ ($\mathrm{PM}_{40}$) to be sampled. The sampled air was dried to a relative humidity below $40\,\%$. An iso-kinetic splitter was used, together with as short as possible tubing, to feed the aerosol to the different instruments inside the measurement container.

The operated low- and high-volume filter samplers were positioned on the upper deck of the RV ($\sim\!28\,\mathrm{m}$ a. s. l.) and each one run on a $\mathrm{PM}_{10}$ inlet. Further details on the filter sampling can be found in subsection 2.3. An ultrasonic anemometer was operated next to the high-volume filter sampler and provided wind direction data for an automatic shut-down mechanism exclusive to the high-volume sampler. Sampling was stopped automatically during periods with wind direction within a half-circle at the sampler centred towards the stack exhaust situated on the RV's stern. A detailed description of the instrument set-up of the ACE-SPACE project is given in Schmale et al. (2019), and a full description of the instrument set-up during ACE is given in the cruise report by Walton and Thomas (2018). In the following, we focus on the instruments used in conjunction with our CCN- and INP-related investigations.

## 2.1 Aerosol Size Distribution

Particle number size distributions (PNSD) of aerosol particles in the mobility diameter range of 11–400 nm and aerodynamic range of 500 nm–19 µm were measured using a Scanning Mobility Particle Sizer (SMPS; custom-built by PSI) and an Aerodynamic Particle Sizer (APS; model 3321 by TSI inc., Shoreview, MN, USA), respectively. The custom-built SMPS instrument is further described in Wiedensohler et al. (2012). Validation of sizing accuracy of both instruments was performed using polystyrene latex spheres. To minimize influences of the ship exhaust, data filtering was performed for the SMPS and APS data, based on sudden changes in the total aerosol particle number concentration, concentrations of carbon dioxide, black carbon, and wind direction (Moallemi et al., 2021).

PNSD of SMPS and APS were merged by assuming spherical particle shape for the SMPS output to convert mobility diameter to geometric diameter as a first step. As a second step, the APS output is converted from aerodynamic to geometric diameter assuming spherical shape and material density of $1.8 \, \mathrm{g \, cm^{-3}}$. The combination of both outputs enables interpolation of the gap between the SMPS ($D_\mathrm{p} \leq 400 \, \mathrm{nm}$) and APS ($D_\mathrm{p} > 500 \, \mathrm{nm}$) instruments. Further, a mode fitting technique analogue to Modini et al. (2015) was applied, that is based on a method described in Khlystov et al. (2004) and fully described in the supplement to Landwehr et al. (2021). In this approach, each PNSD is assumed to be a superposition of up to three aerosol modes and log-normal distributions in pre-defined size ranges are fitted. The three modes are Aitken (modal diameter in the range 1 to 20 nm; referred to as "mode 1" in the following), accumulation (10 to 100 nm; "mode 2"), and a sea-spray mode (centered around 140 to 220 nm; "mode 3"). For each time step, combining all fitted modes results in a smoothed total PNSD that is used where a measured PNSD is too noisy at low concentrations, e.g., for the calculation of the particle hygroscopicity parameter described in subsection 2.2. An example of the mode fitting is given in Fig. S1. Integration of smoothed total PNSD over all geometric diameters gave total aerosol particle number concentration ($N_\mathrm{total}$) for each time step. Analogously, the concentration of particles with $D_\mathrm{p} > 500 \, \mathrm{nm}$ ($N_{>500}$) was derived, which is later used in a commonly-used parameterization for INP concentration (see subsection 3.4).

## 2.2 Cloud Condensation Nuclei

A Cloud Condensation Nuclei counter (CCNc; *CCN-100* instrument by DMT, Longmont, CO, USA) was used to measure the CCN concentration at various $SS$. The CCNc's main part is a continuous-flow thermal gradient diffusion chamber, in which a stream-wise temperature gradient is induced to achieve defined $SS$ and corresponding particle activation to droplets. The aerosol flow rate inside the CCNc is $0.5 \, \mathrm{L \, min^{-1}}$. Activated particles are counted by an optical particle counter. Further documentation on the CCNc can be found in Roberts and Nenes (2005). Calibration of the CCNc was performed prior to the cruise, following the standard operating procedure given in Gysel and Stratmann (2014) and recommendations in Schmale et al. (2017). During ACE, the CCNc was operated at $SS$ of 0.1, 0.15, 0.2, 0.3, 0.5, and $1 \, \%$ maintained for ten minutes each. To ensure stable thermal conditions within the instrument, data collected during the first five minutes of each $SS$ set-point were discarded. Furthermore it was ensured that (1) the instrument's internal thermal stability control reported thermally stable conditions, and (2) the absolute difference between set and read temperature of the optics was smaller than $2 \, \mathrm{K}$. The

remaining data were aggregated into one minute intervals and filtered for ship exhaust influences (same as for the SMPS and APS instruments). Based on the filtered values, averaged $N_{\text{CCN}}$ at a particular $SS$ were calculated. This procedure results in one $N_{\text{CCN}}$ value per hour and supersaturation. During data analysis, CCN concentrations at $0.1\,\%$ were found to lack sufficient data quality, therefore measurements at this supersaturation were discarded.

For determining the critical dry diameters for particle activation ($D_{\text{crit}}$) and aerosol particle hygroscopicity parameters ($\kappa$), we applied the procedure used in, e.g., Kristensen et al. (2016) and Petters and Kreidenweis (2007). $D_{\text{crit}}$ is implicitly defined as the lower boundary of the integral over the PNSD for which the integrated particle number concentration equals the measured CCN number concentration. In our case, the upper boundary of the integral was always $40\,\mu\text{m}$, due to using instruments operated on a PM$_{40}$ inlet. The $\kappa$ value, an indirect measure of chemical composition of the CCN at given $D_{\text{crit}}$, is derived from the $SS$ applied in the CCNc and the corresponding $D_{\text{crit}}$. Corresponding to $N_{\text{CCN}}$, one $\kappa$ value per hour and supersaturation is determined. A Monte Carlo simulation (MCS) approach with an iterative solver was used, following the procedure described in Herenz et al. (2019), to model error propagation in both derivation of $D_{\text{crit}}$ and calculation of $\kappa(D_{\text{crit}})$. The calculation of $D_{\text{crit}}$ and thus $\kappa$ is highly sensitive to the PNSD which, in our case, depends on the quality of the mode-fitting. To exclude unreasonable values, $D_{\text{crit}}$ values were filtered. For this, the range between $10^{\text{th}}$ and $90^{\text{th}}$ percentile of $D_{\text{crit}}$ was calculated for each $SS$ separately. $D_{\text{crit}}$ values outside this range and associated $\kappa$ values were excluded from further analysis. Hence, the presented results are representative of the most frequently occurring $\kappa$ values.

## 2.3 Filter sampling for INP, sodium and MSA analysis

Filter sampling of ambient air for off-line INP, sodium and MSA analysis at the laboratories of Leibniz Institute for Tropospheric Research (TROPOS) was carried out using a high-volume sampler (HV; *DHA-80* filter sampler, DIGITEL, Volketswil, Switzerland). Further, a low-volume sampler (LV; *DPA-14* filter sampler, DIGITEL) was used to collect additional samples for INP analysis. LV sampling was performed at eight hours time resolution using track-etched polycarbonate membrane filters (Whatman Nuclepore, Cytiva, Little Chalfont, UK; $200\,\text{nm}$ pore size, $47\,\text{mm}$ in diameter) at a flow rate of roughly $25\,\text{L}\,\text{min}^{-1}$. The HV sampler used a flow rate of roughly $500\,\text{L}\,\text{min}^{-1}$, sampling air through quartz-fibre filters (*MK 360*, Munktell, Bärenstein, Germany) of $150\,\text{mm}$ in diameter for up to 24 hours per filter. Here, filters showed an average sampled volume of $471.3\pm151.4\,\text{m}^3$ (mean $\pm$ SD) due to individual sampling time ($<1$ to $1437\,\text{min}$) depending on the automatic shutdown mechanism. In total, 258 LV and 94 HV filters were collected throughout the cruise, including five (four) un-sampled reference filters for LV (HV) sampling, called field blank filters (FBF). FBF were handled in the same way as the sampled ones, enabling assessment of background concentrations due to both methodology and handling. After sampling, filters were stored in a freezer at $-20^\circ$C and shipped frozen to TROPOS for off-line analysis after the cruise concluded. INP analysis was performed for both LV and HV filters. LV filters were used solely for the INP analysis, while the HV filters were split between INP, sodium and MSA analysis and reserve samples. HV filters with a too small sampling volume ($<100\,\text{m}^3$), due to the aforementioned automated shut-down mechanism, were not considered further to prevent unreasonably high conversion factors to infer atmospheric concentrations from filter analysis results. A total of 79 sampled HV filters were included in the following analysis.

The immersion freezing capability of the aerosol particles collected on each LV and HV filter was measured using the Ice Nucleation Droplet Array (INDA) at TROPOS. INDA is based on the freezing array method described in Conen et al. (2012) and a detailed instrument description is given in the supporting information in Hartmann et al. (2019). As a first step of the analysis process, stored filters were acclimatised to roughly $-3°C$ in a fridge. LV filter contents were washed off by submerging the filter in $7.5\,\mathrm{mL}$ ($V_{\mathrm{water}}$; $10\,\mathrm{mL}$ at later stages) ultra-pure water (milliQ, $18.2\,\mathrm{M\Omega\,cm^{-2}}$). In contrast, 96 randomly punched-out pieces of $1\,\mathrm{mm}$ in diameter ($D_{\mathrm{punchout}}$) per filter were used for the INP analysis of the HV filters. The 96 wells of a PCR (polymerase chain reaction) plate (BRAND, Wertheim, Germany) were either filled with $50\,\mathrm{\mu L}$ each ($V_{\mathrm{droplet}}$) of the filter washing water (LV) or with $50\,\mathrm{\mu L}$ of milliQ water and one punch-out (HV). The PCR plate was sealed and partially submerged in the ethanol bath of a cryostat (*FP 40*, Julabo, Seelbach, Germany). Cooled at a rate of roughly $1\,\mathrm{K\,min^{-1}}$, the number of frozen droplets ($n_{\mathrm{frozen}}$) and corresponding temperature value ($T$) was documented automatically every six seconds. Recommendations on sample handling and processing given in Polen et al. (2018) were followed.

The frozen fraction ($f_{\mathrm{ice}}$) was calculated by dividing $n_{\mathrm{frozen}}$ by the total number of droplets per PCR plate ($n_{\mathrm{total}} = 96$). Obtained $f_{\mathrm{ice}}$ at any $T$ was used to derive the cumulative INP concentration $N_{\mathrm{INP}}$, using:

$$N_{\mathrm{INP}}(T) = -\frac{ln(1 - f_{\mathrm{ice}}(T))}{V} \tag{1}$$

according to Vali (1971). The reference volume $V$ for the LV filters was calculated as:

$$V = \frac{V_{\mathrm{flow}}}{V_{\mathrm{water}}} * V_{\mathrm{droplet}}, \tag{2}$$

where $V_{\mathrm{flow}}$ is the sampled air volume, $V_{\mathrm{water}}$ is the volume of washing water and $V_{\mathrm{droplet}}$ is the water volume per PCR plate well. For the HV filters, $V$ was calculated using:

$$V = \frac{(0.5 * D_{\mathrm{punchout}})^2}{(0.5 * D_{\mathrm{filter,HV}})^2} * V_{\mathrm{flow}}, \tag{3}$$

where $D_{\mathrm{punchout}}$ is the diameter of the filter sub-sample per well and $D_{\mathrm{filter,HV}}$ the diameter of a HV filter. $V_{\mathrm{flow}}$ was logged by both (LV and HV) samplers. Due to the higher number of LV samples, resulting in more robust statistics compared to the HV samples, we focus in subsection 3.2 on INP results derived from the LV samples, while only briefly commenting on results from the HV samples.

Uncertainties arising from the methodology were assessed similarly to previous studies (e.g., Wex et al., 2019; Gong et al., 2020). Confidence intervals for $f_{\mathrm{ice}}(T)$ of each filter were determined using a method described in Agresti and Coull (1998). Resulting lower and upper values of each $f_{\mathrm{ice}}(T)$ in Eq. 1 are reported as error bars of $N_{\mathrm{INP}}(T)$ values.

Lower and upper limits of INP concentrations are given for cases when ice fractions of $f_{\mathrm{ice}} = 0$ or $f_{\mathrm{ice}} = 1$ were obtained, i.e., none or all wells of the PCR plate were frozen. For these two cases, Eq. 1 is not applicable to calculate concentrations. We then assume the probabilities of either none ($f_{\mathrm{ice}} = 0/96$) or one ($f_{\mathrm{ice}} = 1/96$) of the PCR wells to be frozen as equal. A similar assumption is made in the case of all ($f_{\mathrm{ice}} = 96/96$) or all but one ($f_{\mathrm{ice}} = 95/96$) wells being frozen. Considering $f_{\mathrm{ice}} = 1/96$ and $f_{\mathrm{ice}} = 95/96$ in Eq. 1 yields estimates for the lower and upper limit of detectable $N_{\mathrm{INP}}$, respectively.

Based on the $f_{\mathrm{ice}}$ of the field blank filters (FBF) we determined averaged temperature-dependent INP concentrations, $N_{\mathrm{INP,FBF}}$, which are given as point of reference for background concentration levels whenever $N_{\mathrm{INP}}$ for the sampled filters

are shown. Equation 1 with an average (mean ± SD) volume of sampled air for all sampled LV (HV) filters of $8.95 \pm 0.74\,\mathrm{m}^3$ ($471.3 \pm 151.4\,\mathrm{m}^3$) was used to calculate $N_{\mathrm{INP,FBF}}$. In Tab. S3 $N_{\mathrm{INP,FBF}}$ is given for the LV and the HV filters. No correction for contamination by the RV's stack exhaust was applied, as it was shown in Welti et al. (2020) that ship exhaust is not ice-active in the temperature range we are presenting ($T > -30\,^\circ\mathrm{C}$).

The INP concentrations derived for the LV filters were normalised to the aerosol surface area or alternatively volume,
following Mitts et al. (2021), in order to obtain normalized ice (nucleation) activity. For this, the particle surface area and volume size distributions were first inferred from the number size distribution, assuming spherical particles, and then integrated over the entire diameter range. This was done for each size distribution measurement. These values were averaged over the $8\,\mathrm{h}$ sampling time of each LV filter and $N_{\mathrm{INP}}$ is divided by these values, resulting in the ice active site density ($n_{\mathrm{s}}$) and ice active volume density ($n_{\mathrm{v}}$), respectively.

Analysis of the HV filters regarding mass concentrations of sodium and MSA was performed. Total filter mass load for each HV filter was determined using a micro-balance (*AT261 Delta Range*, Mettler Toledo, Greifensee, Switzerland). Filter contents were extracted and ion chromatography performed, following the procedures described in Müller et al. (2010) and van Pinxteren et al. (2017). Results of the analysis were corrected for standard conditions and are reported as atmospheric mass concentrations (in $\mathrm{\mu g\,m^{-3}}$). An influence from the RV's exhaust stack on the measured sodium or MSA concentrations is not
expected due to their respective marine sources. Sodium is used as a conservative tracer for primary aerosol particles of marine origin, and MSA was found to be solely a product of DMS oxidation (Legrand and Pasteur, 1998).

## 2.4 Further resources

During the ACE cruise sea water was sampled every four hours using the RV's underway water supply system and during CTD (conductivity, temperature and depth) rosette deployments, at specific depths down to $200\,\mathrm{m}$ (Walton and Thomas, 2018).
Glass fibre filters ($25\,\mathrm{mm}$ in diameter, $700\,\mathrm{nm}$ pore size) were sampled with up to $2\,\mathrm{L}$ of sampled sea water under low vacuum pressure and stored at $-80\,^\circ\mathrm{C}$ prior to analysis on-board the RV. After extraction in $90\,\%$ acetone for $24\,\mathrm{h}$, chlorophyll a (Chl-*a*) pigment concentration (in $\mathrm{mg\,m^{-3}}$) was measured on a fluorometer (*AU-10*, Turner Designs, San Jose, CA, USA). Calibration was performed against a standard Chl-*a* solution (Sigma-Aldrich, St. Louis, MO, USA). Concentrations of volatile organic compounds (VOC), like isoprene and DMS, in sea water were measured using a gas chromatography-mass spectrometry system
(*5975-T LTM-GC/MSL*, Agilent Technologies, Santa Clara, CA, USA) by the Surveying Organic Reactive gases and Particles Across the Surface Southern Ocean (SORPASSO) project. A description of the full procedure can be found in Rodriguez-Ros et al. (2020).

Continuous data of wind speed and direction during the cruise were obtained from two ultrasonic anemometers (part of *MAWS 420* system, Vaisala, Vantaa, Finland) located on the port- and starboard side of the RV on the observation deck ($\sim$$30\,\mathrm{m}$
a. s. l.) above the bridge of the RV (Walton and Thomas, 2018). Observed wind speeds were corrected by Landwehr et al. (2020) regarding the instrument's position on the ship. To estimate the wind speed at $10\,\mathrm{m}$ a. s. l. ($U_{10}$), measurement height and atmospheric stability were considered using a logarithmic wind speed profile, including the drag coefficient. Quantification of air-flow distortion bias generated by the RV's structures was performed using the data from the operational ERA-interim

weather model as a free stream reference. The resulting correction was applied to the observed wind speed, leading to a data set of wind speed at $10\,\mathrm{m}$ a. s. l. for the cruise with a five minute time resolution.

The distance between the RV's position and the nearest land for ACE was calculated by Volpi et al. (2020) using the cruise track, coast lines of the continents from the *NaturalEarth* project (version 4.1.0), and additional information on islands in the SO inside the geographic information system *qGIS* (version 3.2.3-Bonn) with the help of the *NNJoin* plugin (version 3.1.2).

Backward-trajectories along the ship track for ACE are available in Thurnherr et al. (2020). Calculations have been performed with the "LAGRANgian analysis TOol" (LAGRANTO). The three-dimensional wind fields used by LAGRANTO are from the six-hourly global operational analyses of ECMWF and short-term forecasts in between the analysis time steps. A variety of variables were interpolated along the trajectories, e.g., the pressure level of the planetary boundary layer (PBL), the condition of the underlying surface (e.g., land, open ocean, ice) or the total precipitation. In our study the air parcel's height in combination with boundary layer height was used to assess when it was within the PBL. When in the PBL, the information on the type of underlying surface was used. The surface below the air parcel was characterised using geographical location information, similar to what is done in Radenz et al. (2021). The full procedure of the analysis is described in the SI to this study.

## 2.5   Correlation analysis

The collected data were used in a correlation analysis. The goal was to characterise the aerosol population on the SO by finding possible connections between their associated quantities, with the strength or lack of correlation as a first hint for potential sources.

Input variables were the MSA and sodium concentrations from the HV filters, INP concentrations at five temperatures ($-8$, $-12$, $-16$, $-20$, and $-24°C$) from the LV filters, $N_{\text{total}}$, $N_{>500}$, particle concentrations of individual PNSD modes ($N_{\text{mode1}}$, $N_{\text{mode2}}$, and $N_{\text{mode3}}$), $N_{\text{CCN}}$ at all measured $SS$ and respective $\kappa$ values, wind speed at $10\,\mathrm{m}$ a. s. l. ($U_{10}$) and in-water Chl-*a* and DMS concentrations. Note that $N_{\text{INP}}$ values from the analysis of the LV filters were considered in the correlation analysis given better statistical robustness due to a larger number of samples. Correlation analysis was performed by calculating Spearman's rank correlation coefficients and associated $p$ values between input variables. As data of diverse temporal resolution were used, the coarsest resolution ($24\,\mathrm{h}$, chemical analysis) was chosen and variables with finer resolution were averaged over 24 hour periods, using arithmetic mean values. For each variable, 79 data points were used for the correlation analysis. This corresponds to the number of HV filters which sampled a sufficient ($>100\,\mathrm{m}^3$) volume (see subsection 2.3).

## 3   Results and Discussion

### 3.1   Aerosol Particles and Cloud Condensation Nuclei

In Fig. 2a, smoothed PNSD for Legs 1–3 of ACE are presented. For the most parts of the cruise, a bi-modal particle number size distribution was present. Potential sources for the pronounced accumulation mode, which is causing the bi-modality, are either

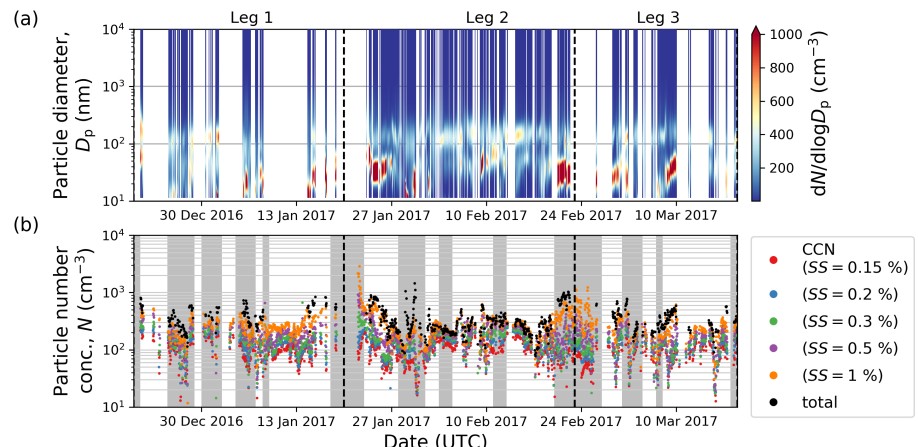

**Figure 2.** Time series of (a) hourly smoothed PNSD and (b) total aerosol particle (black) and CCN number concentration (colour-coded by supersaturation) during Legs 1–3. Ports visited (dashed lines) and vicinity to land (grey area) are indicated in the figure. Data gaps stem from filtering for instrument availability and exclusion of stack exhaust contamination periods. In (a) the Hoppel minimum can be seen as local minimum in the Aitken mode of the PNSD (Leg 1: 48 nm, Leg 2: 74 nm, Leg 3: 68 nm on average).

entrainment of aerosol particles from the free troposphere (FT) into the MBL or in-cloud processing, according to Hoppel et al. (1986). A general characterisation of the aerosol particles sampled during ACE is given in Schmale et al. (2019), including median values for the diameter of the Hoppel minimum, which are 48, 74, and 68 nm for Leg 1, Leg 2, and Leg 3, respectively (see Fig. 2a).

Time series of $N_{\text{total}}$ and $N_{\text{CCN}}(SS)$ for the ACE cruise are given in Fig. 2b. Here, days for which the average distance to land is lower than 200 km are highlighted with grey shading. Additionally, the starts and ends of the different Legs are given as dashed lines. Filtering by stack exhaust contamination caused concurrent data unavailability, while differences in temporal resolution and availability of the instruments create times with no overlap between $N_{\text{total}}$ and $N_{\text{CCN}}(SS)$. Fig. 2b shows that $N_{\text{CCN}}$ at a particular $SS$ varied over 2 orders of magnitude throughout the cruise, e.g., at a $SS$ of 0.2 % ($N_{\text{CCN},0.2}$) from 4 to 309 cm$^{-3}$. In the vicinity of the ports, higher $N_{\text{total}}$ and $N_{\text{CCN}}(SS)$ are observed compared to the open ocean sections (stack exhaust contamination filtering being performed for both as described in subsection 2.2). This suggests aerosol particle abundance to be influenced by terrestrial and anthropogenic sources and is in line with Schmale et al. (2019) showing pristine conditions during ACE being encountered only south of 55° S.

Periodic differences between $N_{\text{total}}$ and $N_{\text{CCN},1.0}$ throughout the cruise were observed (Fig. 2b). Periods of larger differences coincide with PNSD in Fig. 2a featuring a pronounced Aitken mode ($D_{\text{p}} = 10$–100 nm) with elevated numbers in the size range below 40 nm. During these periods even $SS = 1\%$ was not sufficient to activate the smaller Aitken mode particles. Consequently, quantities presented later in this manuscript, that are derived from $N_{\text{CCN},1.0}$, are representative for the larger Aitken mode particles ($D_{\text{crit}}$ at this $SS \sim 30$ nm, see Tab. S1).

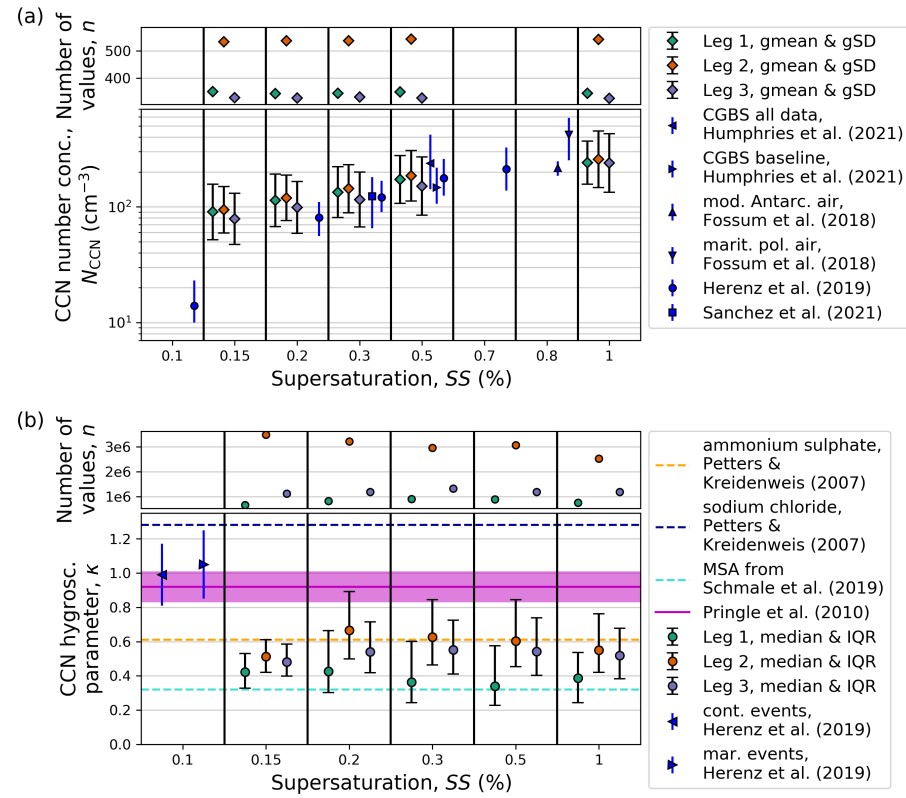

**Figure 3.** (a) Geometric mean values (gmean) and geometric standard deviation (gSD; whiskers) of CCN number concentration ($N_{\text{CCN}}$) and (b) median values and respective inter-quartile range (IQR) of aerosol particle hygroscopicity parameter ($\kappa$). Both $N_{\text{CCN}}$ and $\kappa$ are given as function of supersaturation ($SS$) for Leg 1 (green), Leg 2 (orange), and Leg 3 (purple), respectively. All $\kappa$ values resulting from $D_{\text{crit}}$ values outside of $10^{\text{th}}$ to $90^{\text{th}}$ percentile range for $D_{\text{crit}}$ (per $SS$) are excluded here. Included for comparison in (a) are the averages (median and IQR) over all measurements at Cape Grim Baseline Station (CGBS) coinciding with PEGASO and CAPRICORN-II from Humphries et al. (2021) (triangle pointing left) and measurements for "baseline" conditions (triangle pointing right) during that period. Baseline conditions are defined as wind directions between 190 and $280°$, and ambient radon concentrations below $100\,\text{mBq}\,\text{m}^{-3}$. Averages for events of modified Antarctic air (upward pointing triangle) and maritime polar air (downward pointing triangle) from Fossum et al. (2018) are given. As reference, mean $\kappa$ values for sodium chloride (dashed blue line) and ammonium sulfate (dashed orange line) from Petters and Kreidenweis (2007), and a hypothetical $\kappa$ value for MSA from Schmale et al. (2019) (dashed teal line) are given in (b). Averages for Antarctica from Herenz et al. (2019) for cases of continental (triangle pointing left) and marine events (triangle pointing right) are given. Modelled $\kappa$ values for the Southern Ocean's surface layer (magenta line and area) from Pringle et al. (2010) for reference. The number of data points ($n$) are indicated in the figure.

In addition to the time series presented in Fig. 2b, spatial distribution of $N_{\text{CCN}}$ at all measured $SS$ are given as daily averages in Fig. S2. Averages of $N_{\text{CCN}}(SS)$ for the Legs of ACE are shown in Fig. 3a. Due to the frequency distributions of $N_{\text{CCN}}$ in Fig. 4a (introduced later) resembling log-normal distributions, Leg-aggregated data and variability are given as geometric

mean and geometric standard deviation values, respectively. $N_{\mathrm{CCN}}$ values increase with $SS$, e.g., for Leg 1 from $91\,\mathrm{cm}^{-3}$ ($N_{\mathrm{CCN},0.15}$) to $241\,\mathrm{cm}^{-3}$ ($N_{\mathrm{CCN},1.0}$). For all $SS$, the largest geometric mean values of $N_{\mathrm{CCN}}(SS)$ are observed during Leg 2. Moreover, the average Hoppel minimum diameter was found to be the largest for Leg 2, when compared to Legs 1 and 3, indicating a pronounced Aitken mode. This, together with Schmale et al. (2019) showing less contribution (relative and absolute) of SSA to CCN during Leg 2, suggests a significant fraction of CCN originating from secondary aerosol production. However, differences in $N_{\mathrm{CCN}}$ between Legs are within the ranges given by the respective geometric standard deviations. The longitudinal differences in CCN abundance is either small against the overall variability in the data, or a variety of effects cancel each other out so that no clear longitudinal trend can be observed. A similar conclusion can be drawn in terms of latitudinal trends, because the majority of the cruise track during Leg 2 was south of $60°\,\mathrm{S}$, compared to Legs 1 and 3 being solely north of $60°\,\mathrm{S}$. With this, the CCN concentrations given in Tab. S1 can be considered representative for the MBL over the whole SO region during summertime.

In addition to our data, $N_{\mathrm{CCN}}$ from a selection of other studies performed on Antarctica or over the SO are given in Fig. 3a and summarized in Tab. 1. For the continental Antarctic research station *Princess Elisabeth* (PES), located at $71.95°\,\mathrm{S}$ and $23.35°\,\mathrm{E}$ on East Antarctica's Queen Maud Land and about $200\,\mathrm{km}$ in-land from the Antarctic coast, Herenz et al. (2019) reported $N_{\mathrm{CCN},0.1}$, $N_{\mathrm{CCN},0.2}$, $N_{\mathrm{CCN},0.3}$, $N_{\mathrm{CCN},0.5}$, and $N_{\mathrm{CCN},0.7}$. Overall, we find good agreement between values for the measurement period 2013–2016 in Herenz et al. (2019) and the geometric mean values of this study covering roughly three months during the Austral summer. The reported concentrations of $N_{\mathrm{CCN},0.2}$, $N_{\mathrm{CCN},0.3}$, and $N_{\mathrm{CCN},0.5}$ for cases of maritime air-masses reaching PES show a difference of $-28\,\%$, $-9\,\%$, $+3\,\%$ to our values, respectively. A first hypothetical reason for the differences at $SS = 0.2\,\%$ is that PES is not located directly at the Antarctic coast. A second hypothetical reason is that activation at this low $SS$ is associated with large particles, which might be removed due to atmospheric processes during transport to PES. At the Australian Cape Grim Baseline Station (CGBS; $40.68°\,\mathrm{S}$, $144.68°\,\mathrm{E}$), $N_{\mathrm{CCN},0.5}$ is measured continuously since the mid-1970s (Gras and Keywood, 2017). In Humphries et al. (2021), average $N_{\mathrm{CCN},0.5}$ over ACE's time frame (November 2017 to March 2018) at CGBS are given. For this period, a $N_{\mathrm{CCN},0.5}$ median of $\sim230\,\mathrm{cm}^{-3}$ is reported (triangle pointing left in Fig. 3a). This is above our median value for Leg 1 of $181\,\mathrm{cm}^{-3}$ and at the upper end of our results (IQR: $138\text{–}225\,\mathrm{cm}^{-3}$). Differences could be due to continental air-masses reaching CGBS. Conditions at CGBS are only representative for the SO when the wind direction is between $190°$ and $280°$, the so-called "baseline" conditions (Gras and Keywood, 2017). At CGBS, the ambient radon concentration is used as a proxy for terrestrial influence (e.g., McCluskey et al., 2018a) and a threshold of $100\,\mathrm{mBq\,m}^{-3}$ is used in Humphries et al. (2021). The averaging of the CGBS measurements which feature only baseline conditions gives a median of $\sim130\,\mathrm{cm}^{-3}$ (triangle pointing right in Fig. 3a). This is at the lower end of our results for Leg 1 and we conclude that the terrestrial influence on our $N_{\mathrm{CCN},0.5}$ average values is small. The terrestrial influence during Leg 1 being small is underlined later in the text by the backward trajectory analysis (see subsection 3.2). As for ship-based CCN measurements, comparison between our findings and the PEGASO cruise in the SO's Atlantic sector during January–February 2015 (Fossum et al., 2018) can only be done semi-quantitatively, since $SS$ are not identical. Further, a comparison is only reasonable for Leg 3, the part of ACE on the Atlantic sector of the SO. The result of visual interpolation between our $N_{\mathrm{CCN},0.5}$ and $N_{\mathrm{CCN},1.0}$ for Leg 3 lies in the ranges of $217 \pm 31\,\mathrm{cm}^{-3}$ reported for modified Antarctic air encountered during PEGASO

**Table 1.** Overview of a selection of studies on aerosol particles and CCN over the sumertime Southern Ocean.

| | Location (ACE equivalent) | Time frame | CCN measurements (sampling cut-off) | Reference |
|---|---|---|---|---|
| Princess Elisabeth station | 71.95° S, 23.35° E (Leg 3) | 2013–2016 | $N_{\mathrm{CCN},0.1}$, $N_{\mathrm{CCN},0.2}$, $N_{\mathrm{CCN},0.3}$, $N_{\mathrm{CCN},0.5}$, $N_{\mathrm{CCN},0.7}$ | Herenz et al. (2019) |
| Cape Grim baseline station | 40.68° S, 144.68° E (Leg 1) | Nov 2013– Mar 2016 | $N_{\mathrm{CCN},0.5}$ | Humphries et al. (2021) |
| PEGASO cruise | 50–65° S, 70–40° W (Leg 3) | Jan– Feb 2015 | $N_{\mathrm{CCN},0.8}$ ($PM_{2.5}$) | Fossum et al. (2018) |
| ACE cruise | 34–78° S, circum- Antarctic | Dec 2016– Mar 2017 | $N_{\mathrm{CCN},0.15}$, $N_{\mathrm{CCN},0.2}$, $N_{\mathrm{CCN},0.3}$, $N_{\mathrm{CCN},0.5}$, $N_{\mathrm{CCN},1.0}$ ($PM_{40}$) | This study |
| SOCRATES campaign | 42.5–62.1° S, 133.8–163.1° E (Leg 1) | Jan 2017 | $N_{\mathrm{CCN},0.3}$ | Sanchez et al. (2021) |
| MARCUS cruise | 44–69° S, 60–160° E (Leg 1) | Oct 2017– Mar 2018 | $N_{\mathrm{CCN},0.2}$, $N_{\mathrm{CCN},0.5}$ | Humphries et al. (2021) |
| CAPRICORN-II cruise | 44–68° S, 130–150° E (Leg 1) | Jan– Feb 2018 | $N_{\mathrm{CCN},0.2}$, $N_{\mathrm{CCN},0.3}$, $N_{\mathrm{CCN},0.5}$ | Sanchez et al. (2021), Humphries et al. (2021) |

(Fig. 3a). For the British Southern Ocean (BSO) cruise, only CCN concentrations inferred from nss-sulfate are available in O'Dowd et al. (1997), not comparable with any of our $N_{\mathrm{CCN}}$. As for aircraft-based CCN measurements, $N_{\mathrm{CCN},0.3}$ between 17 and $264\,\mathrm{cm}^{-3}$, with an average of $123 \pm 58\,\mathrm{cm}^{-3}$ (mean $\pm$ SD), are reported in Sanchez et al. (2021) for flights through the MBL between 42.5–62.1° S and 133.8–163.1° E during the Southern Ocean Clouds, Radiation, Aerosol Transport Experimen-

tal Study (SOCRATES). The reported concentrations are slightly lower than what was measured in that area during ACE, with values between 48 and $452\,\mathrm{cm}^{-3}$ and an average of $178 \pm 99\,\mathrm{cm}^{-3}$ (mean $\pm$ SD). Besides the difference in measurement height (SOCRATES: 50 m a. s. l. until height of inversion; ACE: $\sim$15 m a. s. l., see section 2), another factor is that measurements are

from successive years, with the ACE cruise being in that area during 16 January to 26 January 2017 and the 15 flights during SOCRATES in the period of 15 January to 25 February 2018.

An overview on the aerosol particle hygroscopicity parameter $\kappa$ observed during Legs 1–3, is given in Fig. 3b. Leg-wise averages of the hourly available $\kappa$ values per $SS$ are given as median values and respective inter-quartile ranges (IQR) because the frequency distributions of $\kappa(SS)$ in Fig. 4b (introduced later) do not resemble log-normal distributions. Error bars include both natural variability and the measurement uncertainty in $\kappa$, as described in subsection 2.2. Median $\kappa$ values for all Legs and $SS$ are spread between $0.3$ and $0.7$, with a combined variability-uncertainty range (indicated by IQR as error bars) ranging

from $0.2$ to $0.9$. Differences between Legs can be seen, with the highest median values at each $SS$ found for Leg 2. Reference values for pure compounds or compound classes are given in Petters and Kreidenweis (2007), a mean $\kappa$ between $0.1$ and $0.2$ for organic material, $\kappa_{\mathrm{mean}} = 0.6$ for ammonium sulfate, $\kappa_{\mathrm{mean}} = 0.9$ for sulfuric acid, and $\kappa_{\mathrm{mean}} = 1.3$ for sodium chloride is reported. A typical $\kappa$ value for ammonium nitrate is omitted in Fig. 3b, as nitrate-containing compounds were found to not play an important role for the CCN population. Additionally, in Schmale et al. (2019) a $\kappa \sim 0.3$ for MSA is hypothesised. The

majority of our $\kappa$ are above what is given for organic material and below the value given for sulfuric acid, which indicates that the sampled CCN population consists of a variable mixture of organic and inorganic materials. Median $\kappa$ values for Leg 1 are closest to what is assumed for MSA, while median values for Legs 2 and 3 are closer to the value for pure ammonium sulfate. Looking at size-dependency, no clear trend of $\kappa$ between different $SS$ is apparent when considering error bars. For the size range between roughly $30$ and $110\,\mathrm{nm}$ probed by the range of $SS$ (Tab. S1), the chemical composition appears to be

independent of particle size, which further suggests a well-mixed aerosol (or CCN) population. However, when considering median values alone, for $SS > 0.15\,\%$ a slight decrease in $\kappa$ values with increasing $SS$ can be seen for Leg 2. Lower $\kappa$ values at higher $SS$ are in line with condensable organic vapors contributing to the aerosol chemical composition, while larger, aged particles activating at lower $SS$, are associated with higher $\kappa$ (McFiggans et al., 2006). Legs 1 and 3 do not feature increasing $\kappa$ values with decreasing $SS$, which suggests an internally mixed CCN population.

Comparison to Herenz et al. (2019) shows that our $\kappa$ values observed over the SO are much lower than the ones reported for the Austral summer on continental Antarctica, where $\kappa$ values at $SS = 0.1\,\%$ were found to be in the range between $0.8$ and $1.3$. This suggests significantly different particle composition at PES compared to the SO. Herenz et al. (2019) interpreted their sampled particles to be of mostly inorganic nature (i.e., sea salt) and therefore assumed a primary, marine origin. Our values, because of the overall small $\kappa$ values, hint towards a composition dominated by organics. Note that particle sizes between 30

and $110\,\mathrm{nm}$ were probed with the $SS$ range of our instrument, hinting on the encountered aerosol population being mainly comprised of smaller particles. For SOCRATES, Saliba et al. (2020) report $\kappa$ values between $0.2$ and $0.5$ for particles with $D_{\mathrm{p}} < 100\,\mathrm{nm}$. Our $\kappa$ values in this size range (corresponding to $SS > 0.2\,\%$) lie partially in a similar range (Leg 1) and partially on the upper end (Legs 2 and 3) of the range found during SOCRATES, respectively.

    Using a global numerical weather model, simulations of $\kappa$ at $SS = 0.1\,\%$ for the SO region were presented in Pringle et al.

(2010). Model results give values of $0.9 \pm 0.1$ for the surface layer. Comparison to our results suggests an overestimation of sea salt contribution and/or an underestimation of the presence of organic material in the model. A similar effect is noted in Schmale et al. (2019), when the $N_{\mathrm{CCN},0.2}$ measured during ACE are compared to the output of the Global Model of Aerosol

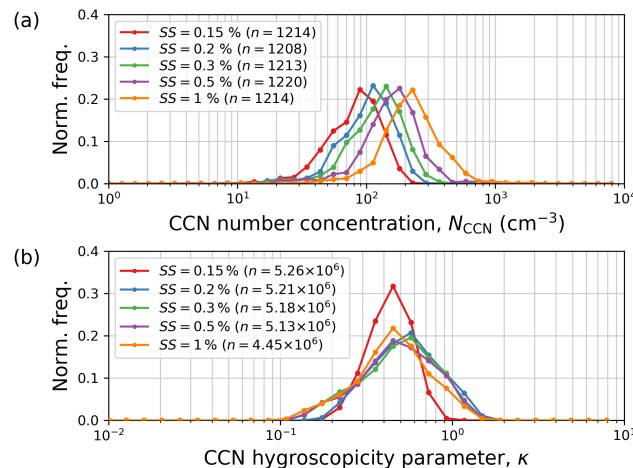

**Figure 4.** Normalized probability density function of (a) CCN number concentration ($N_{\text{CCN}}$) and (b) hygroscopicity parameter ($\kappa$) for levels of supersaturation 0.15, 0.2, 0.3, 0.5, and 1 % ($SS$, colour-coded) of all observations taken during Legs 1–3. The $N_{\text{CCN}}$ result from averaging 5 min long intervals of 1 Hz measurements. Each of the six $SS$-levels is repeated once per hour. The $\kappa$ values result from Monte-Carlo simulation runs ($n_{\text{MCS}} = 10^4$) of hourly smoothed particle number size distributions. All $\kappa$ values that resulted from $D_{\text{crit}}$ outside of $10^{\text{th}}$ to $90^{\text{th}}$ percentile range (per $SS$) are excluded. The number of data points are indicated ($n$) in the figure.

Processes (GLOMAP) model. The largest differences between measurements and model output coincided with the highest gaseous MSA concentrations, suggesting an underestimation of CCN from secondary origin. A global model producing $\kappa$
values for the SO region twice as high compared to what we measured in situ reveals a strong discrepancy in CCN properties and suggests possible model deficiencies in the representation of CCN sources and modelled aerosol-cloud interactions.

    Geometric mean values (and respective geometric standard deviation) of $N_{\text{CCN}}$ and $D_{\text{crit}}$, and median values (and respective IQR) of $\kappa$ for the entire cruise and its Legs are summarized in Tab. S1.

    Probability density functions (PDF) of normalized frequencies for $N_{\text{CCN}}(SS)$ and $\kappa(SS)$ during Legs 1–3 are given in
Fig. 4. PDF of $N_{\text{CCN}}$ (Fig. 4a) show mono-modal distributions for all $SS$, with the PDF maxima shifting towards higher $N_{\text{CCN}}$ with increasing $SS$, e.g., $\sim$90 cm$^{-3}$ at 0.15 % to $\sim$210 cm$^{-3}$ at 1 %. Comparing the distribution for $N_{\text{CCN,0.2}}$ (blue line in Fig. 4a) with yearly-averaged PDF from measurement sites around the globe in Schmale et al. (2018), our values show lower number concentrations with a PDF maximum at $\sim$100 cm$^{-3}$ and share resemblance in terms of number of modes and maximum location with the distribution reported for clean marine conditions (mono-modal, maximum at $\sim$200 cm$^{-3}$). For the
MBL Legs of SOCRATES, the PDF for $N_{\text{CCN,0.3}}$ is bi-modal, with peaks at 100 and 150 cm$^{-3}$ (Sanchez et al., 2021). In their study, the low concentration mode was associated with precipitation events, effectively removing larger particles. The high concentration mode was associated with atmospheric processes causing particle growth, e.g., 1) oxidation of volatile organic compounds and subsequent condensation or 2) cloud processing.

    A change in distribution shape with increasing $SS$ can be seen for PDF of $\kappa(SS)$ in Fig. 4b. All five distributions are
mono-modal, with maxima between 0.4 and 0.6. PDF for $SS$ of 0.3, 0.5, and 1 % (green, purple, and orange line, respectively)

feature a tail towards smaller values of $\kappa$. This occurrence of small particles (activated at high $SS$) consisting of mainly organic material forms a strong case for the sampled Aitken-mode CCN originating from secondary organic aerosol formation and growth processes. The accumulation mode, probed with the measurement at $SS = 0.15\%$, shows similar $\kappa$ values as the Aitken mode (Fig. 4b).

PDF for all $SS$ other than $0.15\%$ feature a tail towards higher $\kappa$ values. Such high $\kappa$ values at high $SS$ seem counter-intuitive and are indicative of highly hygroscopic Aitken mode particles being sampled. A sensitivity study of our methodology with respect to (1) modelling the measurement uncertainty via Monte Carlo simulations (Fig. S3a), (2) consideration of error propagation, and (3) quality of the fitted modes to the PNSD was performed. As $\kappa$ values were robust against these variations, we conclude that this tail (yet counter-intuitive) is not an artefact of our methodology. However, to avoid speculation on the
reason, we take a conservative approach in keeping the focus of the interpretation on the median values presented in Fig. 3b.

### 3.2 Ice Nucleating Particles

Time series of $N_{\mathrm{INP}}(T)$ for $T = -24°C$ ($N_{\mathrm{INP},-24}$; orange), $N_{\mathrm{INP},-20}$ (purple), $N_{\mathrm{INP},-16}$ (blue), $N_{\mathrm{INP},-12}$ (green), and $N_{\mathrm{INP},-8}$ (magenta) are given in Fig. 5a–e, respectively. INP concentrations outside the detectable range (indicated by triangles and estimated as described in subsection 2.3) are represented in the figure by each filter's lowest (lower detection limit) and
highest resolvable concentration value (upper detection limit). For $T \leq -12°C$ (Fig. 5a–d), the respective measurement background INP concentrations are represented via the averaged FBF spectra (dash-dotted lines), as described in subsection 2.3. Measurement uncertainties (indicated by error bars) become smaller with decreasing temperature. This is due to (1) increased freezing probability with decreasing temperature, and (2) the measurement uncertainty being described by binomial sampling confidence intervals (following Agresti and Coull, 1998). The combination of both effects results in smaller error bars at lower
temperatures. At $-12$ and $-16°C$, $N_{\mathrm{INP}}$ show the highest variability of around 3 orders of magnitude. At $-8$ and $-20°C$ the variability decreases to about 2 orders of magnitude, while $N_{\mathrm{INP}}$ at $-24°C$ only varies within 1 order of magnitude. This decrease in the range of values is considered a bias due to $N_{\mathrm{INP}}$ being close to or above the upper detection limit. Each of the shown time series of $N_{\mathrm{INP}}$ in Fig. 5 contains episodes of elevated INP concentrations, coinciding with the RV being close to land (grey area; within $200\,\mathrm{km}$) and harbours (dashed line). During the open ocean sections of the cruise the majority of data
points shows up to 2 orders of magnitude lower concentrations (e.g., $N_{\mathrm{INP},-16} = 0.1\text{–}10\,\mathrm{m}^{-3}$). This suggests that elevated atmospheric INP concentrations are connected to terrestrial (including coastal) INP sources. This assumption is supported by the results of the air-mass origin analysis (subsection 2.4) using the LAGRANTO backward-trajectories for ACE provided in Thurnherr et al. (2020). An overview of the results is given in Fig. S6 showing time series of surface contributions to each LV filter. The time series for the surface type contributions to the PBL signal (Fig. S6c) and the contribution of geographical
regions (Fig. S6d) show that periods of elevated INP concentration (Fig. 5) coincide with periods when air-masses that passed over African, Australian, South American land masses or coastal regions were sampled. Contrary to these regions, air-masses passing over Antarctica did not show higher $N_{\mathrm{INP}}$ than oceanic air-masses (Fig. S7).

    For comparison, Fig. 5c contains the range between the $5^{\mathrm{th}}$ and $95^{\mathrm{th}}$ percentile of $N_{\mathrm{INP},-15}$ from Bigg (1973) (orange area). They sampled filters in the SO around Australia, collecting $0.3$ and $3\,\mathrm{m}^3$ of ambient air through a pair of membrane filters. In

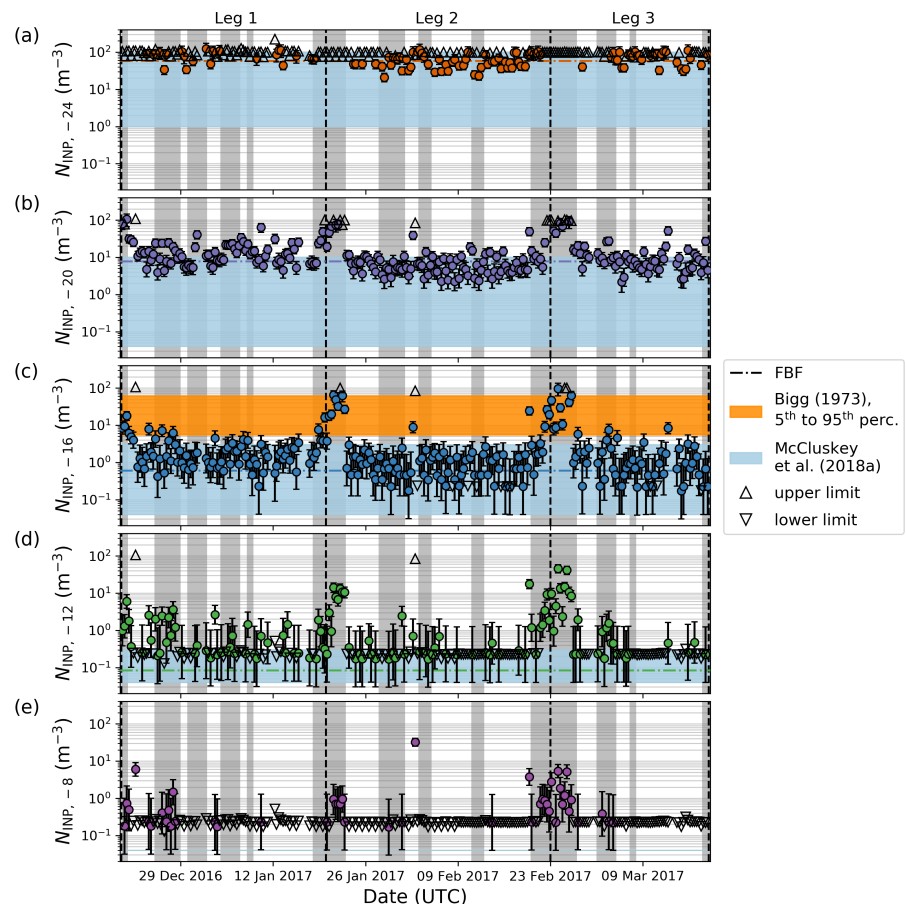

**Figure 5.** Time series of INP number concentration ($N_{\mathrm{INP}}$) at (a) $-24$, (b) $-20$, (c) $-16$, (d) $-12$, and (e) $-8°\mathrm{C}$ from the LV filters sampled during ACE. $N_{\mathrm{INP}}$ values outside the detectable range are indicated by downward (upward) pointing triangles if they are below (above) the lower (upper) edge of the detectable range. The Legs of ACE (dashed lines) and periods when the RV was close to land (grey area) are indicated. In (a–d) the measurement background from averaged spectra of field blank filters (FBF) is indicated (dash-dotted lines). $N_{\mathrm{INP}}$ from McCluskey et al. (2018a) are included for reference (blue area). In addition, $N_{\mathrm{INP},-15}$ from Bigg (1973) are included in (c) as the range between their $5^{\mathrm{th}}$ and $95^{\mathrm{th}}$ percentile (orange area). A correction of the $N_{\mathrm{INP},-15}$ values from Bigg (1973) was applied, following the supporting information to McCluskey et al. (2018a).

terms of sampling strategy, our LV sampling of $12\,\mathrm{m}^3$ through a porous filter over eight hours compares well with the sampling of Bigg (1973). The techniques to measure $N_{\mathrm{INP}}$ were however different. Filters sampled during ACE were analysed with a freezing array method (subsection 2.3), while INP contents in Bigg (1973) were analysed by means of a thermal diffusion chamber. In the SI of McCluskey et al. (2018a), the effect of background INP concentrations during the study of Bigg (1973) is assessed and a correction proposed ($22\,\%$ lower values). This correction was applied to the values shown in Fig. 5c. The

majority of our $N_{\mathrm{INP},-16}$ measurements are in the open ocean sectors and clearly below the range of $N_{\mathrm{INP},-15}$ observed by

Bigg (1973). However, the $N_{\mathrm{INP},-16}$ larger than $10\,\mathrm{m}^{-3}$ at the end of Leg 1, when the RV was in the vicinity of Australia, lie within the range of values given in Bigg (1973).

In McCluskey et al. (2018a) INP measurements from CAPRICORN-I are presented. The range of observed INP concentrations is included in Fig. 5 for comparison. At each temperature, $N_{\mathrm{INP}}$ observed during ACE are at the upper end or higher than concentrations observed during CAPRICORN-I, except at $-16°$C when low concentrations were measured on the open ocean in air-masses without terrestrial influence. Differences in sampled geographical area (CAPRICORN-I: $43$–$53°$ S and $141$–$151°$ E; this study: $34$–$78°$ S, circum-Antarctic) and season (CAPRICORN-I: March–April; this study: December–March) could be reasons for the differences in observed INP abundance. Our results are consistent with preliminary results from MARCUS, CAPRICORN-I & II in McFarquhar et al. (2021), where $N_{\mathrm{INP}}$ in the MBL over the SO are shown to exhibit a large variability, very low overall values and a weak overall latitudinal dependence. Further, the highest concentrations were found near land and values differed largely from historical measurements (e.g., Bigg, 1973). Feedback of the Earth's changing climate on INP in the SO region as a contributor to the observed difference between current and historical observations cannot be ruled out (e.g., Bigg, 1990). However, potential mechanisms behind such a hypothetical feedback have not been identified so far. For completeness, spatial distributions of $N_{\mathrm{INP},-16}$ from our study and $N_{\mathrm{INP},-15}$ from Bigg (1973) are shown in Fig. S4c.

Averaging of the $N_{\mathrm{INP}}$ at selected temperatures has been performed, in order to showcase typical values for the SO region. A summary of average INP concentrations for Legs 1–3 at selected temperatures is given in Tab. S2. Two different approaches were used for averaging the INP concentrations of the LV samples. In the first approach only values which are inside the detectable range are considered and the averages are given as $N_{\mathrm{INP,LV}}$ in Tab. S2. For the second approach, values outside the detectable range ($N_{\mathrm{INP,LV}}^{\star}$) were included by using a value on the edge of the detectable range instead (see subsection 2.3). Results of the two approaches differ in mean, median, and geometric mean concentration values by up to $\pm 50\,\%$. The largest differences were found at a $T$ of $-8$ and $-24°$C, where the number of data points outside the detectable range is largest. We report all averaged values with explicit reference to their potential biases. Average values of $N_{\mathrm{INP},-24}$, $N_{\mathrm{INP},-20}$, $N_{\mathrm{INP},-16}$, $N_{\mathrm{INP},-12}$ and $N_{\mathrm{INP},-8}$, sorted by Legs of ACE, are given in Fig. 6. Values were determined including the estimates for INP concentrations outside the detectable range (triangles in Fig. 5). As a point of reference for the measurement background, concentrations of the averaged FBF are included (dash-dotted lines). Differences in median values between different Legs are largest at $-20°$C, however still within the respective IQR. Mean values (crosses in Fig. 6) are higher than the median, and outside of the IQR for all temperatures other than $-24°$C.

The air-mass origin for the whole cruise and individual Legs are presented in Fig. 7. The average contributions for the whole cruise are dominated by air-masses from the open ocean, with contributions of at least $80\,\%$ and up to $97\,\%$ (Leg 1). The terrestrial air-masses (land and coast; excluding Antarctica) contribute only between $2\,\%$ (Leg 1) and $12\,\%$ (Leg 3). Similar Leg-wise average contributions could, hypothetically, be a result of dominant contribution of "open ocean" conditions during all Legs combined with limited INP variability over the entire SO for "open ocean" conditions.

PDF of $N_{\mathrm{INP}}$ at selected temperatures are shown in Fig. 8. $N_{\mathrm{INP}}(T)$ values outside of the detectable range are not considered for the PDF. As indication for the detectable range, averages for the upper and lower concentration limit are indicated in Fig. 8c–e (dashed line). Interpretation of the PDF for $N_{\mathrm{INP},-8}$ and $N_{\mathrm{INP},-24}$ is omitted due to the low number of samples compared

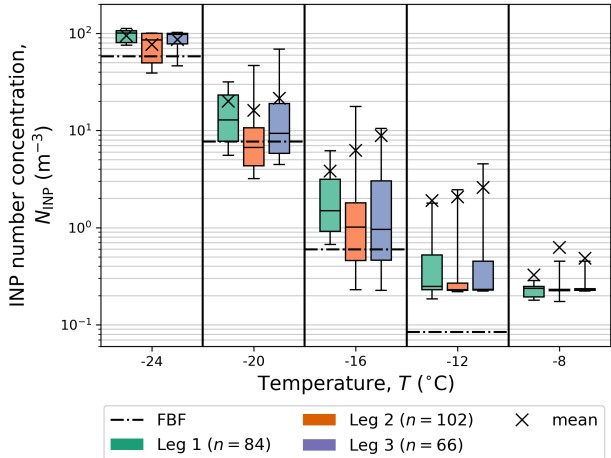

**Figure 6.** Mean values (crosses) and box-and-whiskers plots indicating the median (horizontal lines), inter-quartile range (boxes), and $10^{\text{th}}$ to $90^{\text{th}}$ percentiles (whiskers) of INP number concentration ($N_{\text{INP}}$) from the LV filters sampled during Leg 1 (green), Leg 2 (orange), and Leg 3 (purple). Averaging was performed by treating zero (infinite) values of $N_{\text{INP}}$ at given temperature as values of the lower (upper) limit of the detectable range. In the figure, the measurement background is represented by the averaged spectra of the field blank filters (FBF; dash-dotted lines) and the number of data points ($n$) are indicated.

to other temperatures and concentrations being close to the FBF, respectively. Also for the other temperatures, the overall number of samples considered in the PDF is small. Hence, the following discussion has to be considered semi-quantitative in consequence. The $N_{\text{INP},-12}$ and $N_{\text{INP},-16}$ PDF (Fig. 8c,d) are tri-modal and the $N_{\text{INP},-20}$ PDF (Fig. 8b) is bi-modal. The lowest concentration mode for $N_{\text{INP},-12}$ and $N_{\text{INP},-16}$ contains concentrations below $0.2\,\text{m}^{-3}$ which are on the lower

boundary of the detectable range. Attributing these concentrations to a source or geographical origin is ambiguous when considering the FBF as a point of reference for the background freezing signal. FBF concentrations are $0.08$ and $0.59\,\text{m}^{-3}$ for $-12°\text{C}$ and $-16°\text{C}$, respectively (Tab. S3). We therefore only discuss the two highest concentration modes in the following. The second (first) mode of $N_{\text{INP},-12}$ and $N_{\text{INP},-16}$ ($N_{\text{INP},-20}$), referred to as "low concentration mode" in the following, covers a range of concentration values (e.g., $N_{\text{INP},-16} = 0.3\text{–}10\,\text{m}^{-3}$) encountered mainly during the open ocean sections of

the cruise (Fig. 5b–d). The analysis of the air-mass origin in Fig. 7 shows that air-masses reaching the RV during the sampling of this mode contained mainly open ocean signal ($\sim 90\,\%$) and only small contributions from either Antarctic ($6\,\%$), marine, ice-covered regions (MIZ/sea ice; $2\,\%$), or coastal regions ($1\,\%$). In consequence, we interpret this mode to be dominated by INP of marine origin potentially including some long-range transported terrestrial or coastal INP. This mode is therefore labelled as "open ocean" (light blue area) in Fig. 8b-d. Moallemi et al. (2021) show that fluorescent primary biological aerosol

particles (PBAP) measured during the open ocean sections of ACE originate mainly from SSA. PBAP were found to act as INP in several studies in marine regions of the Northern hemisphere (e.g., McCluskey et al., 2018b; Hartmann et al., 2021) and we assume the same to be the case for the Southern hemisphere. Therefore, we conclude PBAP from SSA to be a potential source for the INP measured on the open ocean sections of the cruise.

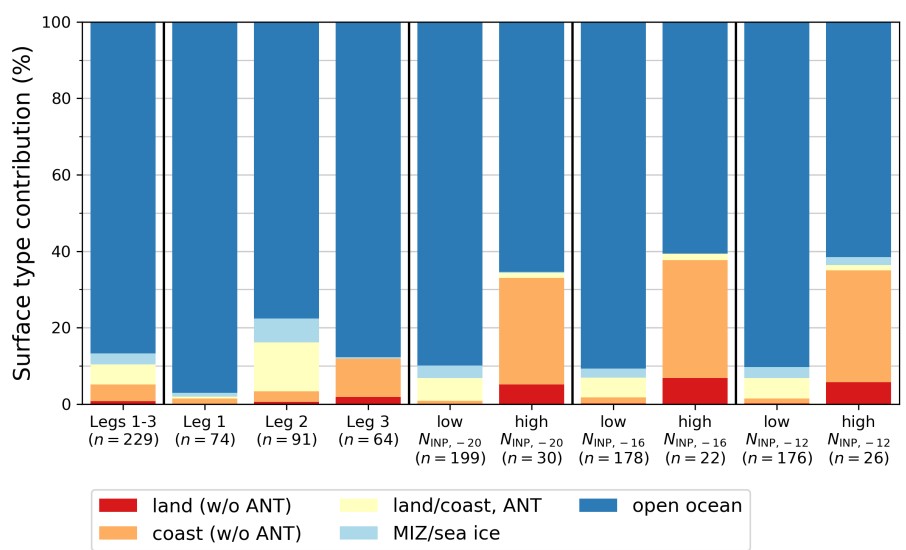

**Figure 7.** Percents of surface type over-passed by 10-day backward-trajectories (see subsection 2.4 and Text S1 in the SI for details). Color codes for surface types are: non-Antarctic land masses (red), non-Antarctic coastal regions or islands (orange), Antarctic continent or coastal regions (ANT; yellow), ice-covered regions (light blue), and open ocean (dark blue). From left to right, the different surface contributions to the air-masses are shown for the entire circumnavigation (Legs 1-3), separated by Leg (see Fig. 1), and cases of $N_{\mathrm{INP},-20}$ below ("low") or above $40\,\mathrm{m}^{-3}$ ("high"), $N_{\mathrm{INP},-16}$ below/above $10\,\mathrm{m}^{-3}$, and $N_{\mathrm{INP},-12}$ below/above $4\,\mathrm{m}^{-3}$ analogous to ranges indicated in the respective PDF (see Fig. 8b–d). The number of trajectory clusters ($n$) are indicated in the figure. Trajectory maps for ACE are available at an hourly resolution from Thurnherr et al. (2020).

The high concentration mode in the PDF of $N_{\mathrm{INP},-12}$, $N_{\mathrm{INP},-16}$, and $N_{\mathrm{INP},-20}$ consists of values (e.g., $N_{\mathrm{INP},-16} = 10$–
$100\,\mathrm{m}^{-3}$) measured close to land (Fig. 5b–d). This mode has a greater terrestrial and coastal influence (combined $\sim 35\,\%$) than the low concentration mode, based on the air-mass origin analysis (Fig. 7). In consequence, the high concentration mode is labelled "terrestrial/coastal" in Fig. 8b–d (yellow area).

In Fig. 8c, a PDF of $N_{\mathrm{INP},-15}$ from Bigg (1973) is included (orange line). Here, only a subset of the total of 126 data points from Bigg (1973) is shown, containing the 58 data points south of $43°\,\mathrm{S}$, mimicking the latitudinal range of the ACE cruise
for a better comparison (Fig. S4). Differences between $N_{\mathrm{INP},-16}$ from this study and $N_{\mathrm{INP},-15}$ from Bigg (1973) are clearly visible, with over 1 order of magnitude lower maximum $N_{\mathrm{INP}}$ observed in our study. Agreement of their values is highest with the subset of our observations in the proximity to land. In a recent study, Cornwell et al. (2020) have shown that re-emission of dust particles from sea water into the atmosphere is possible and that the re-emitted particles retained their ability to act as INP. However, quantifying the contribution of this potential source is not possible with our data set.

Welti et al. (2018) presented PDF of $N_{\mathrm{INP}}(T)$ based on filters sampled at a fixed location on the Cabo Verde island of Sao Vicente, over a four year period (2009–2013). The PDF comprise changes in season, air-mass origin, and bulk aerosol composition. The respective PDF of $N_{\mathrm{INP},-16}$, $N_{\mathrm{INP},-12}$, and $N_{\mathrm{INP},-8}$ are included for comparison in Fig. 8c-e. Welti et al.

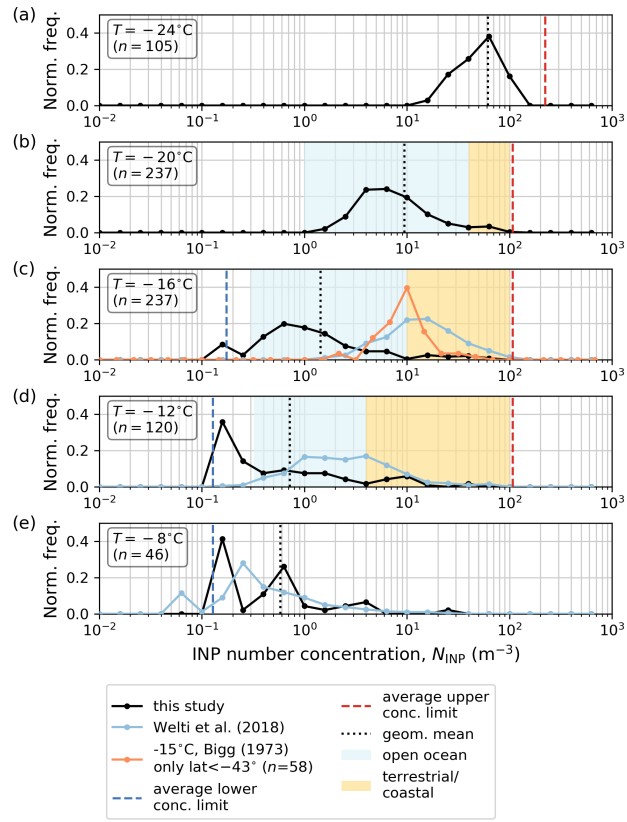

**Figure 8.** Normalized probability density functions (solid black line) and geometric mean values (dotted black line) for INP number concentrations ($N_{\mathrm{INP}}$) at (a) $-24$, (b) $-20$, (c) $-16$, (d) $-12$ and (e) $-8\,^\circ$C from the LV filters sampled during ACE. For reference, $N_{\mathrm{INP},-16}$, $N_{\mathrm{INP},-12}$, and $N_{\mathrm{INP},-8}$ for Cabo Verde (North Atlantic; Welti et al., 2018) are given in (c), (d), and (e), respectively (blue). Additionally, $N_{\mathrm{INP},-15}$ from Bigg (1973) for data points south of $43^\circ$ S (orange) is given for comparison. A classification of modes based on sampling location and air-mass origin is given (yellow area: terrestrial/coastal; light blue area: open ocean). Averages of the upper (dark red) and lower concentration limit (dark blue) are indicated by dashed lines. The number of data points ($n$) are indicated in the figure.

(2018) found log-normal distributions for all temperatures, and attributed them to random dilution during transport, indicating a lack of strong local sources. In other words, the PDF are thought to represent background INP concentrations at the Cabo Verde islands. Comparing PDF, it can be seen that for $N_{\mathrm{INP},-16}$ the bulk of our values is below what is reported in Welti et al. (2018), shifted by roughly 1 order of magnitude towards lower concentrations. For $N_{\mathrm{INP},-12}$ a respective shift is not as clearly seen, however a tendency towards more frequent occurrence of higher concentrations compared to our study is obvious. For $N_{\mathrm{INP},-8}$ this tendency is not visible anymore. The difference between the PDF given in Welti et al. (2018) and our study for $N_{\mathrm{INP},-12}$ and $N_{\mathrm{INP},-16}$ illustrates the latitudinal difference in marine $N_{\mathrm{INP}}$ with lower concentrations in the SO compared to the Atlantic.

The temperature spectra of $N_{\text{INP}}$ for all LV filters sampled during ACE are given in Fig. S8a. The highest freezing onset was found at $-4°C$. Between filters, $N_{\text{INP}}$ is spread over up to 3 orders of magnitude at individual temperatures. This mirrors what can be seen in the PDF in Fig. 8. A typical, steady increase in $N_{\text{INP}}$ with decreasing temperature (1 order of magnitude per 5K) can be observed for the majority of filters and the FBF's curve (pink line). At higher temperatures ($> -20°C$) a shoulder in the

INP spectra of a number of filters is apparent. This feature is unlike the previously mentioned, steady increase in $N_{\text{INP}}$. A high concentration of INP above $-15°C$ is typically associated with a signal of a biological INP source mixed with a mineral or less efficient INP source (e.g., Creamean et al., 2019). In the range of $-12.5$ to $-22.5°C$, only the lowest curves of our spectra lie within the range observed by McCluskey et al. (2018a), who observed even lower $N_{\text{INP}}$ on the SO.

In order to test the hypothesis that typical marine INP (e.g., SSA) were encountered for the majority of the cruise, concen-
tration values were normalised. This was achieved by dividing $N_{\text{INP}}$ by the particle surface area concentration or the particle volume concentration derived from the total aerosol particle number size distributions under the assumption of spherical particles (see subsection 2.3). Normalisation enables comparison of INP properties across different studies, which can include different $N_{\text{INP}}$ derivation approaches or sampled particle size ranges, independent of $N_{\text{INP}}$. The resulting spectra of ice-active number site density, $n_{\text{s}}$, and volume site density, $n_{\text{v}}$, are given in Fig. S8b,c. Values of $n_{\text{s}}$ spread over 4 orders of magnitude
($0.1$–$1000\,\text{cm}^{-2}$) in the observed temperature range. For comparison, values from two laboratory experiments are included in Fig. S8b, that focused on sampling of artificially generated SSA and assessing its ice-activity (DeMott et al., 2016; Mitts et al., 2021). The results from DeMott et al. (2016) span a wider range of $n_{\text{s}}$ and $T$ than the ACE data, while values from Mitts et al. (2021) overlap with the open ocean-sampled filters from ACE. Field measurements from CAPRICORN-I (McCluskey et al., 2018a) showing a large variability in $n_{\text{s}}$, are included for comparison. Contrary to $N_{\text{INP}}$ (Fig. S8a), $n_{\text{s}}$ values from ACE
lie within the lower end of what is reported from CAPRICORN-I. This indicates observation of a similar or more ice-active particle population during CAPRICORN-I compared to ACE. The range of $n_{\text{v}}$ reported in Mitts et al. (2021) are included in Fig. S8c for comparison. The range overlaps with the lower range of the values from ACE. In conclusion, the strong overlap between ice-active site density profiles from ACE (derived from $N_{\text{INP}}$) and studies of artificial SSA (e.g., DeMott et al., 2016; Mitts et al., 2021) supports the idea that low $N_{\text{INP}}$ measured on the open ocean might be driven by SSA.

As mentioned in subsection 2.3, HV filters were also analysed for INP, but due to better higher data coverage (LV: $n_{\text{filter}} = 253$; HV: $n_{\text{filter}} = 79$) we focused on the LV samples. For the sake of completeness we give in Fig. S5 the additional INP spectra determined from HV samples (*DHA-80* sampler, see subsection 2.3). Compared to the LV results in Fig. S8a, the determined $N_{\text{INP}}(T)$ are higher and in a narrower range. LV and HV samples differ in sample collection interval and filter material (LV: poly-carbonate pore filter, $200\,\text{nm}$ pore size; HV: quartz fibre filter). Concerning collection intervals, continuous
sampling over eight hour intervals were chosen for the LV filters to resolve diurnal $N_{\text{INP}}$ variations (non-detected). HV filters were collected during intervals of $24\,\text{h}$, interrupted by breaks due to the automatic shutdown to avoid contamination from ship exhaust. Possible low biases of $N_{\text{INP}}$ for higher sampled volumes have been discussed in Bigg et al. (1963) and Mossop and Thorndike (1966), but do not reflect the trend found for our two sampling techniques. There could be an averaging effect from the longer sampling interval for the HV filters, when sampling from an unevenly distributed INP population. However, such
effects require further investigation. The differences in filter material could be another factor, but are in contrast to Wex et al.

(2020) finding good agreement between quartz fibre and poly-carbonate filters for identical sampling intervals. The higher background INP levels for the HV filters, indicated by higher $N_{\mathrm{INP,FBF}}$ compared to the LV sampling (Tab. S3), hints towards a limited ability of the HV sampling to measure lower INP number concentrations and in consequence overall higher measured INP number concentrations. Contamination from ship exhaust should not effect INP analysis results (see Appendix C in Welti et al., 2020) as exhaust particles are not ice-active in the investigated temperature range. However, deactivation of some INP due to exhaust contamination cannot be ruled out.

### 3.3 Analysis of sodium and MSA

Information on the aerosol chemical composition is widely used to infer the origin of the sampled aerosol particles. To aid the characterisation of CCN and INP sources over the SO, sampled HV filters were analysed regarding the aerosol load and the atmospheric particle mass concentrations of sodium and MSA, two compounds known to be unaffected by stack exhaust.

On average, $32.4\,\mu\mathrm{g\,m^{-3}}$ (median; IQR: $26.1$–$49.6\,\mu\mathrm{g\,m^{-3}}$) of $\mathrm{PM_{10}}$ were observed during ACE (Tab. S5). Leg 1 exhibits a higher median value ($42.4\,\mu\mathrm{g\,m^{-3}}$) compared to Legs 2 and 3 ($31.1$ and $33.3\,\mu\mathrm{g\,m^{-3}}$). Note that contrary to sodium and MSA (see subsection 2.3), an influence of the RV's ship exhaust on $\mathrm{PM_{10}}$ mass cannot be ruled out. However, the quantification of this potential influence is beyond the scope of this study.

Averaging sodium mass concentrations for the whole cruise gives a median value of $2.8\,\mu\mathrm{g\,m^{-3}}$, with an IQR from $1.8$ to $3.9\,\mu\mathrm{g\,m^{-3}}$ (Tab. S5). Higher median values for Legs 1 and 3 compared to Leg 2 are found, similar to what is observed for $\mathrm{PM_{10}}$. This is consistent with Blanchard and Woodcock (1957) showing SSA production to be driven by wave breaking and Schmale et al. (2019) showing on average higher wind speeds and significant wave heights for the Legs with extended open ocean sections (Legs 1 and 3). For the $34^{\mathrm{th}}$ Chinese National Antarctic and Arctic Research Expeditions (CHINARE) cruise on the SO ($40$–$76°$ S, $170°$ E–$110°$ W) in February–March 2018, Yan et al. (2020c) report an average sodium concentration of $0.8 \pm 0.8\,\mu\mathrm{g\,m^{-3}}$ (mean $\pm$ SD). During Leg 2 of ACE, the part of the cruise that has the largest geographical overlap with the region covered during CHINARE, the median sodium mass concentration was $1.8\,\mu\mathrm{g\,m^{-3}}$, i.e., more than two times higher than that observed during CHINARE.

MSA mass concentrations were generally 2 and 1 order of magnitude lower than the ones for $\mathrm{PM_{10}}$ and sodium, respectively. Consequently, values are reported in $\mathrm{ng\,m^{-3}}$ in the following. A median mass concentration of $102\,\mathrm{ng\,m^{-3}}$ for the entire ACE cruise was found (Tab. S5), with highly variable values ranging from $1$ to $455\,\mathrm{ng\,m^{-3}}$. Differences in median values between Legs are very small. The highest concentration of $455\,\mathrm{ng\,m^{-3}}$ was observed on the Ross Sea close to the Antarctic coast (Leg 2). Davison et al. (1996) report for south of the Falkland islands in November 1992 a mean $M_{\mathrm{MSA}}$ of $27\,\mathrm{ng\,m^{-3}}$, with values ranging up to $99\,\mathrm{ng\,m^{-3}}$. During ACE in late February of 2017, values around $120\,\mathrm{ng\,m^{-3}}$ were found in this part of the SO between $70$–$36°$ W. Besides long-term trends over the last two decades, the difference of up to 1 order of magnitude might be due to the difference in season, with higher concentrations for ACE due to increased marine biological activity in early fall compared to late fall for Davison et al. (1996). Another factor is the large degree of variability in MSA abundance across the SO, depending on season and location as illustrated in Castebrunet et al. (2009), with values during ACE on the higher end of the scale. For a number of CHINARE Antarctic cruises, MSA concentrations are reported. Yan et al. (2020b) report for the

polynya regions of the Ross Sea (50–78° S, 160–185° E) an average value of $44 \pm 22\,\mathrm{ng\,m^{-3}}$ (mean $\pm$ SD) for December of 2017 and $39 \pm 28\,\mathrm{ng\,m^{-3}}$ for January 2018. The maximum mass concentration of $211\,\mathrm{ng\,m^{-3}}$ was reported for the Ross Sea, at around $64$–$67°$ S, connected to the position of the dynamic sea ice edge at $\sim64°$ S. Here, with the start of the sea ice melting in early December, the release of iron from ice into the water can spur marine microbial activity (Turner et al., 2004), that may result in an increased DMS emission and consequently secondary MSA production. Consistently, the maximum MSA mass concentration during ACE was encountered near the sea ice edge ($\sim70°$ S) of the Ross Sea in early February 2017. For the Amundsen Sea (40–76° S, 170° E–110° W) in February–March 2018 (34[th] CHINARE cruise), average MSA concentrations of $31 \pm 17\,\mathrm{ng\,m^{-3}}$ are reported in Yan et al. (2020c). The ACE cruise went on the Amundsen Sea in early February 2017 and MSA concentrations in this region show a median value of $210\,\mathrm{ng\,m^{-3}}$. Overall, a difference in average MSA mass concentrations of up to 1 order of magnitude between our study and the CHINARE cruises becomes apparent. One factor might be the usage of different instrumentation and analysis techniques. Another factor causing year-to-year variability could be the presence of sea ice. Schmale et al. (2019) note a significantly lower sea ice extent on the Amundsen Sea during ACE when compared to climatological records. The lack of a sea ice cover enables marine activity and the emission of aerosol precursors into the air. Additionally, variations in atmospheric MSA sink strength are a potential contributor to variability in observed MSA mass concentrations. For example, MSA is efficiently removed from the atmosphere by precipitation. In the SO, rain events are associated with frontal zones. For South Georgia, a sub-micron ($PM_1$) MSA mass of up to $200\,\mathrm{ng\,m^{-3}}$ was reported in Schmale et al. (2013). During ACE, the RV was on station close to this island in the beginning of March 2017, with MSA mass concentrations around $75\,\mathrm{ng\,m^{-3}}$ during these days, underlining the high variability in MSA abundance on the SO.

### 3.4  Correlation Analysis

The results of a correlation analysis performed with a selection of variables gathered during ACE is given as a Spearman rank correlation matrix in Fig. 9.

With regards to the results of our in situ aerosol particle measurement, $N_\mathrm{total}$ was found to be correlated with $N_\mathrm{mode1}$ (correlation coefficient $\rho = 0.9$, $p < .001$) and $N_\mathrm{CCN,1.0}$ ($\rho = 0.8$, $p < .001$). This mirrors the behaviour these quantities show in Fig. 2a, and is indicative for the importance of Aitken mode particles for the total particle and CCN number concentrations at high $SS$.

Correlations between sodium and mode 3 ($\rho = 0.7$, $p < .001$) as well as $PM_{10}$ ($\rho = 0.7$, $p < .001$) concentrations were found. As sodium is used as a conservative tracer for primary aerosol particles of marine origin (Legrand and Pasteur, 1998), especially sea salt, the correlations suggest that SSA significantly contributes to both $PM_{10}$ and the coarse mode. However, we do not find a significant correlation between wind speed ($U_{10}$) and sodium mass concentration. Bates et al. (1998) attributed this kind of observation to the fact that the instantaneous wind speed at the RV is not representative for the conditions an air parcel experienced prior to its measurement. Other studies in the SO region found positive but non-linear connections between wind speed and sodium mass concentrations (e.g., Schmale et al., 2013; Yan et al., 2020a; Landwehr et al., 2021). Another factor might be that the wind speed was averaged over $24\,\mathrm{h}$ in order to match the temporal resolution of the filter sampling. Possible short term effects might be lost due to the averaging process. Note that the wind conditions encountered during ACE

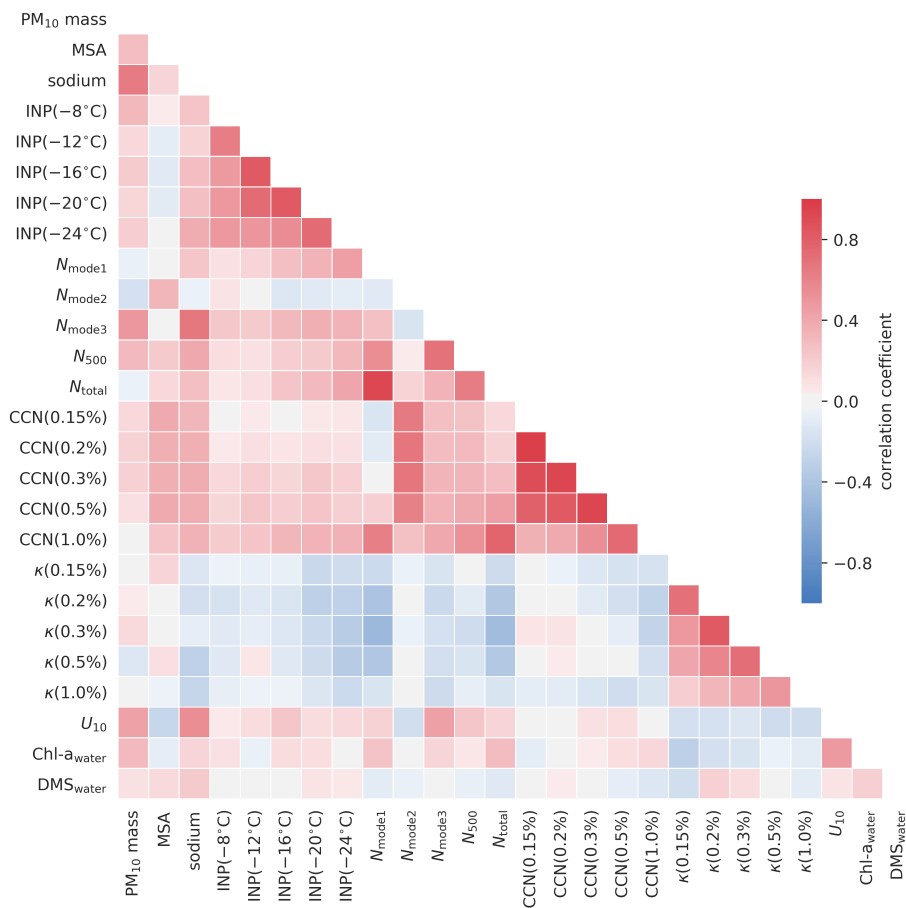

**Figure 9.** Spearman rank correlation matrix of quantities directly measured or derived from measurements during ACE. From the $24\,\mathrm{h}$ long HV sampling, $PM_{10}$ mass and mass concentrations of particulate sodium and MSA are included. INP number concentrations at temperatures of $-8$, $-12$, $-16$, $-20$, and $-24\,°C$ (INP($T$)) are included from the LV filters, sampling for eight hours. Here, the estimates for INP concentrations above and below the detection limit are included. From hourly smoothed particle number size distributions, the total particle number concentration ($N_{\mathrm{total}}$), respective three modes ($N_{\mathrm{mode1}}$, $N_{\mathrm{mode2}}$, and $N_{\mathrm{mode3}}$), and the number concentration of particles larger $500\,\mathrm{nm}$ in diameter ($N_{>500}$) are included. CCN number concentration at 0.15, 0.2, 0.3, 0.5, and $1\,\%$ supersaturation (CCN($SS$)) and derived aerosol particle hygroscopicity parameter ($\kappa(SS)$) are included. Additionally, daily-averaged wind speed at $10\,\mathrm{m\,a.s.l.}$ ($U_{10}$), and in-water concentrations of chlorophyll-a (Chl-$a_{\mathrm{water}}$) and dimethyl sulfide (DMS$_{\mathrm{water}}$) are used. Positive correlation between two quantities indicates a similar trend over time, while an opposing behaviour is indicated by a negative correlation coefficient.

are characterised by median values of 9.88 (Leg 1), 6.62 (Leg 2), and $8.85\,\mathrm{m\,s^{-1}}$ (Leg 3; Schmale et al., 2019) and the relative variability of the daily $U_{10}$ averages is roughly $\pm70\,\%$.

The particle concentration of mode 2 shows a positive correlation ($\rho \approx 0.7$) with $N_{\mathrm{CCN}}$ at $SS \leq 0.5\,\%$, pointing at the importance of accumulation mode particles for the CCN population at atmospherically relevant $SS$. No correlation was found

between $N_{\mathrm{CCN}}$ and mode 3 number concentrations, suggesting little influence of SSA on the CCN population probed with our $SS$.

No correlation between CCN number concentration and MSA mass concentration was found. The lack of a correlation in our analysis seemingly contradicts findings of previous studies (e.g., Ayers and Gras, 1991). This might be a smearing effect due to averaging, since the highest $N_{\mathrm{CCN}}$ (subsection 3.1) coincide with the highest $M_{\mathrm{MSA}}$ (subsection 3.3) when the cruise was in the vicinity of the Antarctic coast (Leg 2). However, finding no correlation does not imply that there could not be a connection under specific conditions and shorter time scales.

Furthermore, no correlation between $N_{\mathrm{CCN}}$ and in-water Chl-$a$ or DMS concentration could be found, which is in line with Ayers et al. (1997). Considering the long process chain from in-water DMS to particles of CCN size, this is not surprising. The argument in Bates et al. (1998), that conditions at measurement point may not be representative for the conditions encountered by the air-parcel during transport, is applicable here as well.

Looking at $\kappa$ values, high correlation between different levels of supersaturation (except $SS = 1\,\%$) could be found, mirroring the lack of size-dependent composition presented in subsection 3.1. Further, no correlation between sodium concentration and $\kappa$ values was found. This indicates that the chemical information for CCN between $30$ and $110\,\mathrm{nm}$ approximated by $\kappa$ is not connected to the mass-dominating, seemingly larger particles represented by sodium and MSA. This again supports the observation of SSA particles not significantly contributing to the CCN population, as SSA dominates the sampled particle mass but not the particle number.

No correlations with any other variable was found for the MSA concentration. This includes the absence of the correlation between MSA and in-water DMS concentration. Although MSA is known to form exclusively from oxidation of DMS in the atmosphere (Sorooshian et al., 2007), a direct correlation is not expected. In-water DMS concentrations are not representative of DMS concentrations in the atmosphere (Ayers et al., 1997) and DMS has an atmospheric lifetime of several days over the SO (Chen et al., 2018).

INP concentrations measured at a temperature difference ($\Delta T$) of $4\,\mathrm{K}$ showed positive correlation ($\rho > 0.6$). This could indicate a common source that contributes INP over a wide $T$-range. For $\Delta T \geq 8\,\mathrm{K}$, only $N_{\mathrm{INP},-12}$ and $N_{\mathrm{INP},-20}$ show a correlation ($\rho = 0.7$, $p < .001$). This correlation between $N_{\mathrm{INP}}$ at $-12$ and $-20°\mathrm{C}$ points at the importance of long-range transport and mixing influencing the INP population in the same way at both temperatures (Welti et al., 2018). The in-water Chl-$a$ concentrations were also included in the correlation analysis, as it can be used as a proxy for biological activity (e.g., McCluskey et al., 2018a). However, no direct correlation between $N_{\mathrm{INP}}$ and Chl-$a$ was found. This suggests that the measured INP are not originating from local biogenic sources but does not exclude a time-shifted response. In DeMott et al. (2010) a parameterization for $N_{\mathrm{INP}}$ is given that is based on $N_{>500}$ for terrestrial conditions. We find no respective correlation between our measured $N_{\mathrm{INP}}$ and $N_{>500}$, underlining that a parameterization based solely on $N_{>500}$ is not applicable for a marine environment as the SO region. A number of additional INP parameterisations are available in the literature that include the normalisation of the INP concentration to the particle surface (e.g., McCluskey et al., 2018a) or volume (e.g., Mitts et al., 2021). Both approaches of normalisation were performed with the ACE data (Fig. S8b,c) and show good agreement with previous studies of marine environments. However, as discussed below the absence of a correlation between $N_{\mathrm{INP}}$ and neither $\mathrm{PM}_{10}$ nor

$N_\text{total}$ shows that both denominators (particle volume or surface) for the normalisation are not directly linked to the INP. No significant correlations were found between $N_\text{INP}$ and $PM_{10}$, $N_\text{total}$ or any other physical and chemical properties measured. This clearly shows that deriving INP-related properties from total number-based or total mass-based aerosol properties without considering air-mass history might lead to results not representative for atmospheric $N_\text{INP}$. Instead, INP concentrations must be compared with results from methods selective to individual, rare particle types that could act as INP and the data must be segregated in terms of air-mass properties or some other, more specific INP tracers such as mineral dust or proteins, to further elucidate INP sources.

## 4 Summary

During the Austral summer of 2016/17, we performed in situ measurements of $PM_{40}$ and filter sampling of $PM_{10}$ aerosol particles for characterizing the physical and chemical properties of aerosol particles over the Southern Ocean during the Antarctic Circumnavigation Expedition. We focused on the abundance and properties of CCN and INP. An air-mass origin and a correlation analysis was performed to identify CCN and INP sources and interpret possible links between different aerosol physico-chemical parameters.

For the in situ measured aerosol particles, bi-modal aerosol particle number size distributions (PNSD) with a distinct Hoppel minimum between 50 and 80 nm were found (Fig. 2a). When the RV was close to continental land-masses (including Antarctica), increased total particle ($N_\text{total}$) and CCN number concentrations ($N_\text{CCN}$) were observed (Fig. 2b). The absolute difference between $N_\text{total}$ and $N_\text{CCN}$ varied during the cruise and was associated with particle activation in the Aitken mode size range. This indicates an importance of the Aitken mode for the CCN population and cloud-formation. Generally, $N_\text{CCN}$ spanned 2 orders of magnitude (e.g., at $SS = 0.3\,\%$ from roughly 3 to 590 cm$^{-3}$), with the respective probability density functions (PDF) sharing resemblance with distributions in Schmale et al. (2018) for clean marine conditions of other locations around the globe. Averages of $N_\text{CCN}$ per Leg (Fig. 3a) showed little difference between the Legs and compare well ($<30\,\%$ percentage difference) with measurements of previous studies in the SO region. Values of the aerosol hygroscopicity parameter $\kappa$ were found to be in the range between 0.2 and 0.9, corresponding to mixtures with different amounts of organic and inorganic materials. Our $\kappa$ are about a factor of two lower than what was measured, e.g., over continental Antarctica or modelled for the SO region. Average values of $\kappa$ were found to be independent of $SS$ and thus particle size (Fig. 3b), indicating in first approximation an internally mixed CCN population in the Aitken and accumulation modes. The PDF of $\kappa$ values was found to be mono-modal for $SS = 0.15\,\%$ (Fig. 4c), while for higher $SS$ tails towards smaller $\kappa$ values were found, hinting at an increasing amount of organics in the smaller Aitken mode particles. In addition, tails towards higher $\kappa$ at $SS > 0.15\,\%$ indicate the occurrence of highly hygroscopic Aitken mode particles. The correlation analysis showed little-to-no connection between the CCN number concentration and quantities from the offline filter analysis, e.g., the concentrations of sodium and MSA (Fig. 9). This is due to the fact that the in situ measured aerosol properties considered here are governed by particle number, while the quantities determined from the filter samples (except for INP) are governed by particle mass. This often implies a focus on different size ranges. However, a connection was found through a positive correlation between total particle number

concentration of the coarse mode and sodium mass concentration (Fig. 9). In addition, the absence of correlation between the sodium mass concentration and CCN number concentration clearly implies that SSA is not an important source of CCN. This agrees well with previous findings, e.g., in Schmale et al. (2019).

Analysis of filter-collected atmospheric aerosol samples for $N_{\mathrm{INP}}$ yielded temperature dependent concentrations between $-4$ and $-27^\circ$C (Fig. S8a). Typically, the $N_{\mathrm{INP}}$ from one filter sample increased by 3 orders of magnitude within steps of $-10^\circ$C. Time series of $N_{\mathrm{INP}}$ showed elevated values coinciding with the RV being in the proximity of land (Fig. 5). This points towards terrestrial and/or coastal sources influencing the INP population. The air-mass origin (Fig. S6) underlines this finding. The comparison with other studies shows that the concentrations observed during ACE are lower than what was observed previously over the SO (Bigg, 1973), while being on the upper end of what is reported in McCluskey et al. (2018a) for a specific sector of the SO in March–April 2016 (Fig. 5 and Fig. S8a). The PDF for $N_{\mathrm{INP}}(T)$ shows two concentration modes at $-20$, $-16$, and $-12^\circ$C (Fig. 8). The analysis of backward-trajectories indicates that low concentrations are associated with air-masses from the open ocean and from Antarctica, while the air-masses transporting higher $N_{\mathrm{INP}}$ passed over (non-Antarctic) land. INP spectra (Fig. S8a) for the most part of the cruise feature a steady increase in INP concentration with decreasing temperature. Features in the spectra of increased $N_{\mathrm{INP}}$ at warm temperatures indicate warm-temperature INP which are connected to biological origin (e.g., Creamean et al., 2019). The correlation analysis indicates correlations between $N_{\mathrm{INP}}$ in the temperature range between $-12$ and $-24^\circ$C. We interpret this signal as indication that mixed long-range-transported populations of INP of biogenic origin ($T > -20^\circ$C) and mineral dust ($T < -20^\circ$C) were present. Indications for local INP sources are very rare, and no correlation between $N_{\mathrm{INP}}$ and Chl-$a$, as proxy for biological activity in the ocean, was found.

The results for the analysis of sodium and MSA in the sampled $PM_{10}$ show that during ACE we encountered (mass-wise) a marine aerosol environment with typical SSA signals. Sodium concentrations showed a median of $2.8\,\mu\mathrm{g\,m}^{-3}$ (Tab. S5). A moderate positive correlation between sodium and $PM_{10}$ (Fig. 9) underlined the importance of SSA for the sampled mass. During ACE, MSA concentrations were found to be highly variable, with a median of $102\,\mathrm{ng\,m}^{-3}$ (Tab. S5). Values were up to 1 order of magnitude higher than in comparable studies and seasonal variation seems to be one reason. The location of peak MSA concentrations near the sea ice edge is consistent with other studies. Similar patterns in the occurrence of maximum MSA concentrations and the hypothesised $\kappa$ value for MSA were found. However, a clear connection between MSA and CCN concentrations or $\kappa$ values did not show in our correlation analysis. With our data covering all sectors of the SO and the rich variety of atmospheric conditions encountered during the cruise, we conclude that such a connection might only be event-based.

The presented data set gives a unique, circum-Antarctic view on CCN and INP abundance, their properties and indications towards aerosol particle origin. Our data give insights into the conditions on the SO regarding cloud-relevant aerosol particles, compare well with previous studies and found already use in climate modelling (Regayre et al., 2020) and remote sensing applications (Efraim et al., 2020).

*Data availability.* The ACE data are available through web portal ZENODO, with INP data from LV sampling under https://zenodo.org/record/4311665 (version 1.1), CCN data under https://zenodo.org/record/4415495 (version 1.1), and data on MSA and sodium under https:

//zenodo.org/record/3922147 (version 1.0). INP data from HV sampling is available from the authors upon request. Further data sets are available at relevant citations within the manuscript.

*Author contributions.* CT performed the analysis and interpretation with contributions from SH, AW, JS, and FS. CT, SH, and FS wrote the manuscript. SH, AW, AB, JS, and MH performed the measurements during ACE. MGB and FS provided the in situ instrumentation. MvP provided the analysis of sodium and MSA of the filter samples. RM provided the PNSD data. All authors contributed to the writing and review of the manuscript.

*Competing interests.* The authors declare no competing interest.

*Acknowledgements.* ACE was a scientific expedition carried out under the auspices of the Swiss Polar Institute, supported by funding from the ACE Foundation and Ferring Pharmaceuticals. This work was supported by the Deutsche Forschungsgemeinschaft (DFG) in the framework of the priority programme "Antarctic Research with comparative investigations in the Arctic sea ice areas" SPP 1158 (grant STR 453/12-1). EU FP7 project "BACCHUS" (project number 603445) is acknowledged for financial support. Julia Schmale holds the Ingvar Kamrad Chair for Extreme Environments Research. Andrea Baccarini was supported by the Swiss National Science Foundation grant No. 169090. The authors would like to thank the PIs of ACE's project 1, D. Antoine and S. Thomalla, for the chlorophyll a data. All data processing was performed using *Python* (version 2.7.14) on *Ipython* (version 5.4.1; Pérez and Granger, 2007). The correlation analysis was made possible by the *spearmanr* function of the *scipy.stats* package for *Python*. All figures in this study were created using the *Matplotlib* package for *Python* (Hunter, 2007).

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
