# Peer review of "Circum-Antarctic abundance and properties of CCN and INP"

_Atmospheric Chemistry and Physics, 2021_

## Author Response (AR1)

**Summary**

We thank the two reviewers for their positive and constructive feedback. Two major changes to the revised manuscript are:

- We included the results of an analysis of the air-mass origin for the sampling of the LV filters (following RC2.3). A full description of the analysis is given in Text S1 in the supporting information (SI). A time series is presented in Fig. S6. Average contributions are presented in Fig. 7. $N_{INP}$ as a function of terrestrial contributions to the air-mass origin are presented in Fig. S7.
- We included a normalisation of the $N_{INP}$ values to the total aerosol particle surface area and volume (following RC2.4). The resulting number site density and volume site density spectra are presented in Fig. S8b and Fig. S8c, respectively.

RC = Reviewer Comment, AC = Author Comment, new suggested text in blue

Note, line numbers refer to the original manuscript, the location in the revised can be seen best in the tracked changes version document. We indicate these line numbers in ().

**Answers to Reviewer 1**

Anonymous Referee #1, 25 Oct 2021

Summary

**RC1.1:** Tatzelt et al. present an extensive data set of CCN and INP measurements from the Antarctic Circumnavigation Expedition of 2016/2017. I understand this to be a very useful addition to the literature, as little data has been available for the Southern Ocean region so far. The article is well-written and contains well-elaborated figures that adequately convey the important features of the data set. The reader is well guided through the discussion, which is fairly detailed but avoids over-interpretation. I recommend publication in ACP, and only have a couple of specific comments and technical corrections as listed below.

**AC1.1:** We thank the reviewer for the positive and very constructive review. Their comments made our study more targeted, structured, and understandable. We have rewritten the text following the major and general comments below.

Specific comments

**RC1.2:** P4, L100-102: The introduction into the present study is rather brief. I think it would be fair to already mention here that parts of the data set have already been presented in Schmale et al. (2019) and Welti et al. (2020) (only mentioned later in line 416) and to summarize the main conclusions from these studies. Schmale et al. (2019) e.g. already presented some analysis on the role of sea spray for CCN. And then underline that you present here the full data set with all the CCN measurements at different supersaturation levels and the temperature-dependent INP records, as well as the correlation analysis to find potential links between the measured properties.

**AC1.2:** We now mention the overview paper for ACE and highlight the added value of the current study. The text was changed accordingly (L109 tracked changes).

Parts of the CCN and INP data set presented in this study have previously been presented in the overview on the ACE cruise in Schmale et al. (2019). Aerosol properties were found to be highly heterogeneous over the SO. The CCN abundance in the MBL showed a significant sea spray contribution in the strong westerly wind belt, while in the polynyas of the Ross and Amundsen Sea biogenic emissions are more important. INP abundance was shown to be lower on the SO than in Northern hemispheric marine air, with small differences between samples on the open ocean and close to the Antarctic coast. INP abundance was found to be similar to other studies on the SO (e.g., McCluskey et al., 2018a) but lower than historic data from Bigg (1973).
INP concentrations at −15°C from the ACE expedition have been presented in Welti et al. (2020). They show that $N_{INP}$ from ship-based measurements are lowest in polar regions and highest in temperate climate zones. Overall, geographical variation in $N_{INP}$ is below 2 orders of magnitude at any temperature. At low temperatures, lower $N_{INP}$ were encountered in the Southern hemisphere than in the Northern hemisphere and this was attributed to the concentration of dust particles active as INP.
These two previous studies (Schmale et al., 2019; Welti et al., 2020) presented a subset of the CCN and the INP data in larger contexts. This paper focuses on the detailed analysis and interpretation of the observations, including $N_{CCN}$ at all available $SS$ and $N_{INP}$ at the full investigated temperature range. Based on 10-day backward-trajectories, an air-mass analysis was performed to locate potential INP sources. In addition, a correlation analysis was performed using CCN, INP, and additional data from the ACE expedition in order to find potential links between the measured properties.

**RC1.3:** P9, L256: You may already include here a brief explanation that the LV filters allow for more robust statistics and were therefore included in the correlation analysis, and that also the overall interpretation of the INP results is mainly based on the LV filters. The HV-filter INP results are not mentioned for a long time (up to line 480) and you can state the reason here.

**AC1.3:** We inserted the following sentence to explain why the LV filter results were chosen for the analysis (L232 tracked changes).

Due to the higher number of LV samples, resulting in more robust statistics compared to the HV samples, we focus in subsection 3.2 on INP results derived from the LV samples, while only briefly commenting on results from the HV samples.

**RC1.4:** P15, L408: Could you please also indicate the range of the McCluskey et al. (2018) INP results in Fig. 5 (similar to the Biggs data)? These data are referred to rather late and only very briefly in the discussion (L476 ff, Fig. 8). I think the comparison with McCluskey et al. (2018) deserves a more prominent place in the discussion already in the context of Fig. 5.

**AC1.4:** The data range of CAPRICORN-I (McCluskey et al., 2018a) has been added to Fig. 5. Further, the results for CAPRICORN-II and MARCUS (McFarquhar et al., 2021) have been added to the discussion (see below).

- L486 tracked changes: In McCluskey et al. (2018a) INP measurements from CAPRICORN-I are presented. The range of observed INP concentrations is included in Fig. 5 for comparison. At each temperature, $N_{INP}$ observed during ACE are at the upper end or higher than concentrations observed during CAPRICORN-I, except at −16°C and −20°C, when low concentrations were measured on the open ocean in air-masses without terrestrial influence.

- L491 tracked changes: Our results are consistent with preliminary results from MARCUS, CAPRICORN-I & II in McFarquhar et al. (2021), where $N_{\mathrm{INP}}$ in the MBL over the SO are shown to exhibit a large variability, very low overall values and a weak overall latitudinal dependence.

References: McFarquhar et al. (2021), doi: 10.1175/BAMS-D-20-0132.1

**RC1.5:** P16, L432: In particular for N_INP(-16°C), the low concentration mode with 0.2 m-3 is not only close to the detectable range, but is also only a small addition on top of the mean INP concentration of the blank filters at that temperature (0.59 m-3, Table S3). I doubt that it is justified to interpret it as an individual mode. Also the shape of the PDFs for -20 and -24°C could be strongly influenced by the background, as can be seen be the strong overlap of the measurement data with FBF ± 2*FBF in Fig. 8. This should at least be mentioned in the discussion.

**AC1.5:** The text has been expanded with the following part to emphasise that there is no point in over-interpreting the third, lowest concentration mode. In addition, the interpretation of –24°C was avoided due to the closeness to the FBF.

- L524 tracked changes: Interpretation of the PDF for $N_{\mathrm{INP},-8}$ and $N_{\mathrm{INP},-24}$ is omitted due to the low number of samples compared to other temperatures and concentrations being close to the FBF, respectively.
- L431 (L529 tracked changes): The lowest concentration mode for $N_{\mathrm{INP},-12}$ and $N_{\mathrm{INP},-16}$ include concentrations below 0.2 m$^{-3}$ which are on the lower boundary of the detectable range. Attributing these concentrations to a source of geographical origin is ambiguous when considering the FBF as a point of reference for the background freezing signal. FBF concentrations are 0.08 and 0.59 m$^{-3}$ for −12°C and −16°C, respectively (Tab. S3). We therefore only discuss the two highest concentration modes in the following.

Minor comments and technical corrections:

**RC1.6:** P1, L6: Add comma after "and".

**AC1.6:** The sentence has been edited as suggested (L5, L5 tracked changes).

An overview of CCN and INP concentrations on the Southern Ocean is provided and, using additional quantities, insights regarding possible CCN and INP sources and origins are presented.

**RC1.7:** P1, L10: You frequently quote the numbers 37 − 123 nm for the average critical dry diameters in the text (also e.g. L345, L356), but the geometric mean values in Table S1 are rather from about 30 − 110 nm.

**AC1.7:** Thank you for pointing this out. The text has been updated for consistency (see below).

- L9 (L9 tracked changes): No distinct size-dependence of $\kappa$ was apparent, indicating homogeneous composition across sizes (critical dry diameter on average between 30 and 110 nm).

- L284 (L329 tracked changes): Consequently, quantities presented later in this manuscript, that are derived from $N_{\text{CCN},1.0}$, are representative for the larger Aitken mode particles ($D_{\text{crit}}$ at this $SS \sim 30$ nm, see Tab. S1).
- L345 (L380 tracked changes): For the size range between roughly 30 and 110 nm probed by the range of $SS$ (Tab. S1), the chemical composition appears to be independent of particle size, which further suggests a well-mixed aerosol (or CCN) population.
- L575 (L717 tracked changes): This indicates that the chemical information for CCN between 30 and 110 nm approximated by $\kappa$ is not connected to the mass-dominating, seemingly larger particles represented by sodium and MSA.

RC1.8: P2, L45: You may note here that this process is called immersion freezing, because you use this term later in line 82.

AC1.8: We now mention immersion freezing at this point of the manuscript (L189, L212 tracked changes).

The immersion freezing capability of the aerosol particles collected on each LV and HV filter was measured using the Ice Nucleation Droplet Array (INDA) at TROPOS.

RC1.9: P4, L93: This is just one example where you sometimes switch from past to present tense when describing older studies, this should be homogenized.

AC1.9: The text was accordingly homogenized (L93, L100 tracked changes).

Additionally, Uetake et al. (2020) showed that bacteria sampled during CAPRICORN-II are mostly of marine origin, suggesting a restricted meridional transport of continental aerosol towards the SO.

RC1.10: P5, L136: Maybe replace "Thereby" by "In this approach"

AC1.10: The text has been updated as suggested (L136, L158 tracked changes).

In this approach, each PNSD is assumed to be a superposition of up to three aerosol modes and log-normal distributions in pre-defined size ranges are fitted.

RC1.11: P6, L142: Shouldn't it be "all geometric diameters"?

AC1.11: Thank you for your comment. Indeed, diameters have been converted from mobility to geometric diameter, as mentioned earlier in this paragraph. The text has been changed accordingly (L141, L163 tracked changes).

Integration of smoothed total PNSD over all geometric diameters gave total aerosol particle number concentration ($N_{\text{total}}$) for each time step.

RC1.12: P10, second line of figure caption: dashed instead of dotted lines

AC1.12: The caption of Fig. 2 has been changed accordingly (P10, P11 tracked changes).

Ports visited (dashed lines) and vicinity to land (grey area) are indicated in the figure.

RC1.13: P10, L287: There are occasional sentences with lots of insets and commas that are difficult to read. Here you could just delete both commas and also delete "individually".

**AC1.13:** The text has been revised in an attempt to simplify longer sentences. This sentence has been updated (L287, L332 tracked changes).

Averages of $N_{CCN}(SS)$ for the Legs of ACE are shown in Fig. 3a.

**RC1.14:** P11, L292: As mentioned above, previous findings from Schmale et al. (2019) could also briefly be discussed in the introduction.

**AC1.14:** Please see AC1.2

**RC1.15:** Fig. 3: You could update Sanchez et al. (2020) to (2021) in the legend.

**AC1.15:** The reference in Fig. 3a has been updated accordingly.

**RC1.16:** P12, L296: You still discuss here that there is no clear longitudinal trend, don't you?

**AC1.16:** Yes, this sentence is dedicated to longitudinal observations. We changed the sentence to clarify this (L296, L340 tracked changes).

The longitudinal differences in CCN abundance is either small against the overall variability in the data, or a variety of effects cancel each other out so that no clear longitudinal trend can be observed.

**RC1.17:** P13, L331: Delete the inset ", individually,".

**AC1.17:** The insert has been removed (L331, L383 tracked changes).

An overview on the aerosol particle hygroscopicity parameter $\kappa$ observed during Legs 1–3, is given in Fig. 3b.

**RC1.18:** P13, L343: Please homogenize the spelling of "Leg" with a first capital letter. (there are numerous instances)

**AC1.18:** We corrected the spelling of "leg" and "legs" to "Leg" and "Legs" throughout the text and figures.

**RC1.19:** P14, second line of figure caption: What is meant here by a "five minute resolution"? Does it refer to the measurement period per supersaturation set point?

**AC1.19:** We averaged 5 minute intervals of 1 Hz measurements. Each of the six $SS$-levels was run for 10 minutes and the first 5 minutes were discarded to ensure thermodynamic equilibrium in the CCNc. $SS$-settings were repeated once each hour. The caption of Fig. 4 has been changed to make this clearer (P14, P17 tracked changes).

The $N_{CCN}$ result from averaging 5 min long intervals of 1 Hz measurements. Each of the six $SS$-levels is repeated once per hour.

**RC1.20:** P15, L406/407: Please reformulate the sentence starting with "However ..." It is rather complicated. You could make a full stop after "land" and start a new sentence with, e.g.: "It does not account for the actual air mass origin and thus is not ..."

**AC1.20:** The part elaborating the methodology in Moallemi et al. (2021) has been removed (L404, L464 tracked changes).

**RC1.21:** P15, L409: The inset ", respectively" could also be deleted.

**AC1.21:** The insert has been omitted (L408, L475 tracked changes).

They sampled filters in the SO around Australia, collecting 0.3 and 3 m³ of ambient air through a pair of membrane filters.

**RC1.22:** P16, L432: "exhibits" and "contains"

**AC1.22:** The sentence has been reformulated (L431, L529 tracked changes).

The lowest concentration mode for $N_{\mathrm{INP},-12}$ and $N_{\mathrm{INP},-16}$ contains concentrations below 0.2 m⁻³ which are on the lower boundary of the detectable range.

**RC1.23:** P17, L452: "dust particles"

**AC1.23:** Following the suggestion, the text has been updated (L452, L561 tracked changes).

In a recent study, Cornwell et al. (2020) have shown that re-emission of dust particles from sea water into the atmosphere is possible and that the re-emitted particles retained their ability to act as INP.

**RC1.24:** P17, L457: "can not" -> "cannot" (there are other instances)

**AC1.24:** This sentence has been removed (L455, L565 tracked changes). However, another part of the text has been updated accordingly (L502, L634 tracked changes).

Note that contrary to sodium and MSA (see subsection 2.2), an influence of the RV's ship exhaust on $PM_{10}$ mass cannot be ruled out.

**RC1.25:** P17, L462: "presented", see above.

**AC1.25:** Please see AC1.9

RC1.26: P17, L464: "Fig.7c-e"

**AC1.26:** We changed the reference accordingly (L464, L573 tracked changes).

The respective PDF of $N_{\mathrm{INP},-16}$, $N_{\mathrm{INP},-12}$, and $N_{\mathrm{INP},-8}$ are included for comparison in Fig. 8c-e.

**RC1.27:** P18, L492-494: As noted above, this argument could already be mentioned earlier in the manuscript.

**AC1.27:** Please see AC1.3

**RC1.28:** P19, L536: Maybe: "Another factor causing year-to-year variability could be the presence of sea ice."

**AC1.28:** The text has been changed following the suggestion (L536, L670 tracked changes).

Another factor causing year-to-year variability could be the presence of sea ice.

RC1.29: P19, L537: Delete comma after "ACE".

**AC1.29:** The text has been updated accordingly (L536, L670 tracked changes).

Schmale et al. (2019) note a significantly lower sea ice extent on the Amundsen Sea during ACE when compared to climatological records.

RC1.30: P19, L538-540: This sentence is also nested and reads a bit complicated. Please reformulate.

**AC1.30:** The sentence was reformulated to improve readability (L538, L674 tracked changes).

Additionally, variations in atmospheric MSA sink strength are a potential contributor to variability in observed MSA mass concentrations. For example, MSA is efficiently removed from the atmosphere by precipitation. In the SO, rain events are associated with frontal zones.

RC1.31: P20, L553: Delete comma after "both".

**AC1.31:** The text has been updated accordingly (L552, L688 tracked changes).

As sodium is used as a conservative tracer for primary aerosol particles of marine origin (Legrand and Pasteur, 1998), especially sea salt, the correlations suggest that SSA significantly contributes to both $PM_{10}$ and the coarse mode.

RC1.32: P20, L556: Delete comma after "mentioning"

**AC1.32:** This part has been completely revised (L556, L692 tracked changes).

Other studies in the SO region found positive but non-linear connections between wind speed and sodium mass concentrations (e.g., Schmale et al., 2013; Yan et al., 2020a; Landwehr et al., 2021).

RC1.33: P20, L557: Delete both commas in the sentence starting with "Another factor"

**AC1.33:** The text has been changed accordingly (L557, L694 tracked changes).

Another factor might be that the wind speed was averaged over 24 h in order to match the temporal resolution of the filter sampling.

RC1.34: P20, L559: Delete comma after "Note"

**AC1.34:** This part has been completely revised (L559, L696 tracked changes).

Note that the wind conditions encountered during ACE are characterised by median values of 9.88 (Leg 1), 6.62 (Leg 2), and 8.85 m s$^{-1}$ (Leg 3; Schmale et al., 2019) and the relative variability of the daily $U_{10}$ averages is roughly ±70 %.

RC1.35: P20, L562: "atmospherically" relevant

**AC1.35:** The text has been updated accordingly (L561, L699 tracked changes).

The particle concentration of mode 2 shows a positive correlation ($\rho \sim 0.7$) with $N_{CCN}$ at $SS \leq 0.5$ %, pointing at the importance of accumulation mode particles for the CCN population at atmospherically relevant $SS$.

RC1.36: P20, L571-573: Again a rather long sentence that could be split into two parts.

**AC1.36:** The sentence was split to improve readability (L571, L711 tracked changes).

Considering the long process chain from in-water DMS to particles of CCN size, this is not surprising. The argument in Bates et al. (1998), that conditions at measurement point may not be representative for the conditions encountered by the air-parcel during transport, is applicable here as well.

RC1.37: P21, L594: Maybe better "total number-based or total mass-based"

**AC1.37:** The sentence was changed accordingly (L594, L742 tracked changes).

This clearly shows that deriving INP-related properties from total number-based or total mass-based aerosol properties without considering air-mass history might lead to results not representative for atmospheric $N_{INP}$.

RC1.38: P22, L625/626: See above, the findings from Schmale et al. (2019) (aren't they also based on a subset of the ACE CCN data?) could already be shortly addressed in the introduction.

**AC1.38:** Please see AC1.2

RC1.39: P22, L647: "a unique"

**AC1.39:** The sentence was changed accordingly (L647, L801 tracked changes).

The presented data set gives a unique, circum-Antarctic view on CCN and INP abundance, their properties and indications towards aerosol particle origin.

RC1.40: P22, L651: "through"

**AC1.40:** The text has been updated accordingly (L651, L805 tracked changes).

The ACE data are available through web portal ZENODO, with INP data from LV sampling under https://zenodo.org/record/4311665 (version 1.1), CCN data under https://zenodo.org/record/4415495 (version 1.1), and data on MSA and sodium under https://zenodo.org/record/3922147 (version 1.0).

RC1.41: P33, last but one line: "are" indicated

**AC1.41:** The caption of Fig. 8 has been updated, accordingly (P33, P23 tracked changes).

The number of data points ($n$) are indicated in the figure.

RC1.42: Caption of Fig. S4: Wrong order of the panels; duplicate "the" in the last but one line.

**AC1.42:** The caption of Fig. S4 has been updated accordingly (see below).

INP number concentration at (a) −24, (b) −20, (c) −16, (d) −12, and (e) −8°C for the LV filters sampled during ACE (circles).

**RC1.43:** Caption of Fig. S5: Replace "and corresponding factor two" by "± a factor of two" as in Figure 8.

**AC1.43:** The caption of Fig. S5 has been updated as suggested (see below).

Average spectra of field blank filters (FBF) ± a factor of two (pink line and area) and data range from McCluskey et al. (2018) (light blue envelope) are given for reference.

**RC1.44:** Table S2: What do you mean by "during parts" of the cruise? No Leg-resolved data are shown. You could also better describe the two different approaches for averaging as in lines 422-424 of the article.

**AC1.44:** Thank you for your comment. The caption of Tab. S2 has been updated to clarify the differences in averaging (see below).

Overview of the LV sampling INP number concentrations ($N_{\text{INP,LV}}$) measured throughout the cruise, given as mean, median, and geometric mean values (including one geometric standard deviation).

**RC1.45:** Table S4: See above, what do mean by "during parts"?

**AC1.45:** The caption for Tab. S4 has been updated to clarify (see below).

Overview of the HV sampling INP number concentrations ($N_{\text{INP,HV}}$) encountered throughout the cruise, given as mean, median, and geometric mean values (including one geometric standard deviation). Averaging was performed with the inclusion of concentrations on the lower/upper boundaries of sensitivity and the number of samples are indicated ($n$) in the table. Differences in sampling strategy for HV compared to LV samples can be found in subsection 2.3.

**RC1.46:** Table S5: You could chose "M" as symbol for the mass concentration.

**AC1.46:** This is a good idea. The mass concentration values have been denoted throughout the text by *M*.

**Answers to Reviewer 2**

Anonymous Referee #2, 17 Dec 2021

Summary

**RC2.1:** Tatzelt et al. present results from detailed measurements of CCN and INP over the Southern High Latitudes during the Antarctic Circumnavigation Expedition (ACE). These measurements are extremely valuable to the field of polar aerosol-cloud interactions. The authors include a discussion of how these measurements compare with previous measurements of CCN and INPs and also perform correlation analyses with co-located aerosol metrics in aims of determining the source origin of measured CCN and INP abundances and variability. Overall, there is massive value in these measurements and the manuscript is well written with few typos. However, there are many studies in the literature that have not be considered in the interpretation of the INP data that I think will

add significant scientific value to the discussion. Additionally, I found several statements that were not adequately supported with data. As such, I recommend publication in ACP after the authors address the major concerns identified below.

**AC2.1:** We thank the reviewer for the positive and constructive review. Their comments made our study much more targeted, structured, and understandable. We have therefore rewritten the text following the major and general comments below.

Major Comments:

RC2.2: The reported INP number concentrations were compared to the Bigg 1973 measurements. However, more recent measurements reported by McCluskey et al., (2018) indicated up to 100 times lower INPs compared to the Bigg 1973 survey. Many, perhaps the majority, of INP number concentrations in this study are significantly higher than those reported by McCluskey et al. (2018) and more recently McFarquhar et al. (2021), which is interesting. The more recent data is only briefly mentioned and discussed in comparison to the Bigg survey. These ACE measurements should be discussed in the context of both Bigg (1973) and McCluskey et al. (2018) datasets for completeness.

**AC2.2:** Please see AC1.4. We have now included a detailed comparison to McCluskey et al. (2018).

RC2.3: The authors indicate that an assessment of air mass origin is not possible for the ACE observations due to a lack of a tracer measurement (lines 403-404). Given the difference in these measurements and those reported in McCluskey et al. (2018) and more recently McFarquhar et al. (2021), it would be extremely valuable to determine if the air masses originated from open ocean waters, the Antarctic coast, or one of the surrounding land regions. While dust concentrations are extremely low over the Southern Ocean, it is widely recognized that the ice nucleation ability of dust is significantly greater than sea spray aerosol (DeMott et al., 2016) and so even small amounts can significantly influence the measured INP. Dust sources have been identified for Antarctica (e.g., Neff et al., 2015). Have the authors considered using something like HYSPLIT back trajectories to determine air mass origin?

**AC2.3:** Backward-trajectories for ACE are available in Thurnherr et al. (2020, https://zenodo.org/record/4031705) and we now include an analysis of the air-mass origin in comparison to $N_{INP}$ from LV filter samples. We added a detailed description of the methods to the SI. Results are shown in the new Fig. 7. Overall, this analysis helped to underline our findings regarding the origin of the two distinct modes in the PDF for $N_{INP,-12}$, $N_{INP,-16}$, and $N_{INP,-20}$ in Fig. 8b— d. However, there is no clear signal in INP abundance from air-masses passing over Antarctica (Fig. S7b).

- L465 tracked changes: This assumption is supported by the results of the air-mass origin analysis (subsection 2.4) using the LAGRANTO backward trajectories for ACE provided in Thurnherr et al. (2020). An overview of the results is given in Fig. S6 showing time series of surface contributions to each LV filter. The time series for the surface type contributions to the PBL signal (Fig. S6c) and the contribution of geographical regions (Fig. S6d) show that periods of elevated INP concentration (Fig. 5) coincide with periods when air-masses that passed over African, Australian, South American land masses or coastal regions were sampled. Contrary to these regions, air-masses passing over Antarctica did not show higher $N_{INP}$ than oceanic air-masses (Fig. S7).
- L517 tracked changes: The air-mass origin for the whole cruise and individual Legs are presented in Fig. 7. The average contributions for the whole cruise are dominated by airmasses from the open ocean, with contributions of at least 80 % and up to 97 % (Leg 1). The terrestrial air-masses (land and coast; excluding Antarctica) contribute only between 2 % (Leg 1) and 12 % (Leg 3). Similar Leg-wise average contributions could, hypothetically, be a result of dominant contributions of "open ocean" conditions during all Legs combined with limited INP variability over the entire SO for "open ocean" conditions.

- L551 tracked changes: This mode has a greater terrestrial and coastal influence (combined ∼35 %) than the low concentration mode, based in the air-mass origin analysis (Fig. 7). In consequence, the high concentration mode is labelled "terrestrial/coastal" in Fig. 8b–d (yellow area).
- L782 tracked changes: The PDF for $N_{\text{INP}}$ (T ) shows two concentration modes at −20, −16, and −12°C (Fig. 8). The analysis of backward-trajectories indicates that low concentrations are associated with air-masses from the open ocean and from Antarctica, while the air-masses transporting higher $N_{\text{INP}}$ passed over (non-Antarctic) land.

**RC2.4:** The authors reference the DeMott et al. (2010) INP parameterization briefly. However, studies (e.g., Vergara-Temprado et al., 2017) have found limitations of DeMott et al. (2010) for marine regions due to the fact that the data used to develop the DeMott et al. (2010) parameterization did not include marine data. Why have the authors not included a comparison against existing marine INP parameterizations based on total aerosol surface area (McCluskey et al., 2018) aerosol volume (Mitts et al., 2021), which could be tested against the measured aerosol distributions from the merged size distributions?

**AC2.4:** References to these parameterisations are now included. Number site density and volume site density have been calculated and added as Fig. S8b and Fig. S8c. The text has been changed, accordingly (see below).

- L595 tracked changes: In order to test the hypothesis that typical marine INP (e.g., SSA) were encountered for the majority of the cruise, concentration values were normalised. This was achieved by dividing $N_{\text{INP}}$ by the particle surface area concentration or the particle volume concentration derived from the total aerosol particle number size distributions under the assumption of spherical particles (see subsection 2.3). Normalisation enables comparison of INP properties across different studies, which can include different approaches to INP number derivation, independent of $N_{\text{INP}}$. The resulting spectra of ice-active number site density, $n_{\text{s}}$, and volume site density, $v_{\text{s}}$, are given in Fig. S8b,c. Values of $n_{\text{s}}$ spread over 4 orders of magnitude ($10^{-1}$–$10^3$ cm$^{-2}$) in the observed temperature range. For comparison, values from two laboratory experiments are included in Fig. S8b, that focused on sampling of artificially generated SSA and assessing its ice-activity (DeMott et al., 2016; Mitts et al., 2021). The results from DeMott et al. (2016) span a wider range of $n_{\text{s}}$ and $T$ than the ACE data, while values from Mitts et al. (2021) overlap with the open ocean-sampled filters from ACE. Field measurements from CAPRICORN-I (McCluskey et al., 2018a) showing a large variability in $n_{\text{s}}$, are included for comparison. Contrary to $N_{\text{INP}}$ (Fig. S8a), $n_{\text{s}}$ values from ACE lie within the lower end of what is reported from CAPRICORN-I. This indicates observation of a similar or more ice-active particle population during CAPRICORN-I compared to ACE. The range of $v_{\text{s}}$ reported in Mitts et al. (2021) are included in Fig. S8c for comparison. The range overlaps with the lower range of the values from ACE. In conclusion, the strong overlap between ice-active site density profiles from ACE (derived from $N_{\text{INP}}$) and studies of artificial SSA (e.g., DeMott et al., 2016; Mitts et al., 2021) supports the idea that low $N_{\text{INP}}$ measured on the open ocean might be driven by SSA.
- L736 tracked changes: A number of additional INP parameterisations are available in the literature that include the normalisation of the INP concentration to the particle surface (e.g., McCluskey et al., 2018a) or volume (e.g., Mitts et al., 2021). Both approaches of normalisation were performed with the ACE data (Fig. S8b,c) and show good agreement

with previous studies of global marine environments. However, as discussed below the absence of a correlation between $N_{INP}$ and neither $PM_{10}$ nor $N_{total}$ shows that both denominators (particle volume or surface) for the normalisation are directly linked to the INP.

References: Mitts et al. (2021), doi: 10.1029/2020GL089633

RC2.5: Why are values above or below the INP detection limit included in figures and analysis? I find this very misleading and distracting. By including the below (above) detection limit values, the impression will be that the INP number concentrations are higher (lower) than what they may actually be. Because you are unable to measure, these values should simple be excluded (NaN).

**AC2.5:** There are two options: 1) to exclude the values on the border of the detectable range or 2) to include these values. To our understanding, both approaches cause a bias in the averaging of the considered values. We addressed this issue in the text (L480–486, L612–620 tracked changes) and provide average values for both cases in Tab. S2. This gives the potential user the option to choose the best version of the data set fit for purpose.

Introduction

RC2.6: Lines 83-85 - I do not believe the CAPRICORN II INP measurements have been publish other than in the McFarquhar et al. (2021) overview paper; McCluskey et al. (2018) only includes CAPRICORN I. (see also Lines 90-92)

**AC2.6:** Thank you, this is correct. The text has been changed accordingly, to reflect that McCluskey et al. (2018a) reported the CAPRICORN I cruise and data from CAPRICORN II is presented in McFarquhar et al. (2021).

- L83 (L88 tracked changes): Two recent cruises: the Cloud, Aerosols, Precipitation, Radiation and Atmospheric Composition campaign (CAPRICORN-I & II). For CAPRICORN-I, observed $N_{INP}$ over the SO in the temperature range between –12 and –31°C varying between 0.04 and 1000 m$^{-3}$ (McCluskey et al., 2018a).
- L91 tracked changes: Preliminary INP results for CAPRICORN-II are presented in McFarquhar et al. (2021) and underline the findings for CAPRICORN-I of low but highly variable $N_{INP}$ values on the SO.

RC2.7: Note that Bigg also hypothesized a decade decline in southern ocean INP concentrations (see Bigg 1990).

**AC2.7:** The reference has been included as hypothesis in addition to the differences arising from the different methods used to derive INP number concentrations (L493 tracked changes).

Further, the highest concentrations were found near land and values differed largely from historical measurements (e.g., Bigg, 1973). Feedback of the Earth's changing climate on INP in the SO region as a contributor to the observed difference between current and historical observations cannot be ruled out (e.g., Bigg, 1990). However, potential mechanisms behind such a hypothetical feedback have not been identified either, tot he best of our knowledge.

RC2.8: Line 95: "In consequence, a dominance of sea spray on INP was concluded." - this should specify that the Southern Ocean *marine boundary layer* was dominated by sea spray INPs.

**AC2.8:** The sentence has been edited as suggested (L95, L102 tracked changes).

In consequence, a dominance of sea spray on the INP population in the SO's MBL was concluded.

RC2.9: Figure 1 - Can the locations of the key ports and locations be added for the reader to more easily follow along?

**AC2.9:** Fig. 1 has been updated to include markers and labels indicating the locations of Cape Town, Hobart, and Punta Arenas (P4, P5 tracked changes).

Methods

RC2.10: What size range is measured with the CCN instrument and how does this compare to the PM10 ionic composition measurements used in the correlation analysis? Can the authors provide some insight into the size ranges when introducing the correlation analysis in Section 3.4?

**AC2.10:** The CCN counter sampled ambient air through a $PM_{40}$ inlet (shared by multiple instruments in the measurement container) and the HV sampled ambient air through a $PM_{10}$ inlet (sampler-exclusive). The usage of these particular inlets defines the upper limit of the sampled particle size range to 40 µm and 10 µm, respectively. With the help of the size distributions, we calculated the respective critical dry diameter to the CCN concentration value (subsection 2.2) and found, on average, values between 30 and 100 nm (Tab. S1), depending on the *SS*. These critical dry diameter values are the lower limit of the particle size range sampled by the CCNc. The lower limit of the HV sampler depends on the filter material. In our case, filters consist of quartz fibres (MK 360 by Munktell) and have a particle retention of 99.9995% at around 300 nm (according to test method ASTM D 2986-91). However, giving the sampled size range is less important than the fact that number and mass dominated quantities are correlated. This is addressed in the text (L575–577, L717–719 tracked changes).

RC2.11: Line 180 - "sampling time (<1 to 1437 min) dependent on the automatic shut-down mechanism." - are the sampling volumes available or perhaps a mean/median sampling volume with standard deviation/IQR would help inform the reader of the typical sampling volume, since this statement suggests the sampling time ranged from 1 minute to ~24 hours.

**AC2.11:** The mean and one standard deviation of the sampled volume have been added for clarity (L179, L201 tracked changes).

Here, filters showed an average sampled volume of 471.3 ± 151.4 m³ (mean ± SD) due to individual sampling times (<1 to 1437 min) depending on the automatic shut-down mechanism.

RC2.12: Lines 176 - was the automatic shut-down mechanisms used for both LV and HV? Please clarify.

**AC2.12:** Indeed, the text was not clear enough that the automatic shut-down mechanism was only used for the HV sampling and has been changed accordingly (L116, L138 tracked changes).

An ultrasonic anemometer was operated next to the high-volume filter sampler and provided wind direction data for an automatic shut-down mechanism exclusive to the high-volume sampler.

**RC2.13:** Line 233 - should this be "specific depths down to 200 m" ?

**AC2.13:** The formulation has been changed as suggested (L232, L264 tracked changes).

During the ACE cruise sea water was sampled every four hours using the RV's underway water supply system and during CTD (conductivity, temperature and depth) rosette deployments, at specific depths down to 200 m (Walton and Thomas, 2018).

Results

**RC2.14:** Figure 2 - The caption mentions data were removed for instrument availability and ship exhaust filtering. There are many instances where CCN data are available, but the size distribution measurements are missing; are all of these times when the SMPS & APS were down?

**AC2.14:** Yes. The same data filter for the ship exhaust was applied to the CCN data as for the SMPS and APS data. Meanwhile, the instrument availability for the CCN instrumentation is independent of the availability of the SMPS and APS instruments.

**RC2.15:** Figure 3 - It's quite challenging to see the differ points. Suggest to widen the size of this figure such that the space between the supersaturation bins are wider?

**AC2.15:** Fig. 3 has been updated to address this issue and improve readability (P11, P13 tracked changes).

**RC2.16:** Line 296 - ".. cancel each other out so that no clear latitudinal" - should this be longitudinal?

**AC2.16:** Yes, please see AC1.16.

**RC2.17:** Line 307 - Can the authors expand on the second reason for differences (number of measurements during Austral summer is higher (three years)? Why would this lead one to expect differences?

**AC2.17:** Thank you for your question. We were referring to a potential effect of the inter-annual variability. However, we cannot think of good reasons actually. It might be the location of the station and the methodology. The text has been re-formulated accordingly (L306, L353 tracked changes).

A first hypothetical reason for the differences at $SS = 0.2$ % is that PES is not located directly at the Antarctic coast.

**RC2.18:** Line 308 - I found the wording of this confusing; suggest change to "Disagreement of CCN concentrations at lower SS between Herenz et al. (2019) and this study suggests that differences arise from larger particles that are typical of activation at lower SS" ?

**AC2.18:** The sentence has been changed as suggested (L307, L355 tracked changes).

A second hypothetical reason is that activation at this low $SS$ is associated with large particles, which might be removed due to atmospheric processes during transport to PES.

**RC2.19:** Line 310-316 - I think the authors are trying to describe two sets of data that are

included in Figure 3, but I think the labels may be inconsistent with the text - The "all data" I think refers to the first portion of this discussion and the "baseline" data refers to the later. However, the text refers to these as "baseline" (Line 313) and then mentions baseline values removed but reference the triangle right (line 316), which is labelled as "CGBS baseline" in Figure 3. Can the authors clarify?

**AC2.19:** The wording was not specific enough to emphasize that the later includes only measurements during "baseline" conditions. The text has been re-phrased accordingly (L315, L365 tracked changes).

Averaging CGBS measurements which feature only baseline conditions, a median of ~130 cm³ (triangle pointing right in Fig. 3a) was found.

RC2.20: Line 317 - "We conclude that terrestrial influence on our average values is small" should specify average values of *CCN_0.5*, since data are unavailable for other SS.

**AC2.20:** "$N_{CCN,0.5}$" has been added for clarity as suggested (L316, L366 tracked changes).

This is at the lower end of our results for Leg 1 and we conclude that the terrestrial influence on our $N_{CCN,0.5}$ average values is small.

RC2.21: Line 322 – suggest to include definition of BSO again here.

**AC2.21:** The text has been updated accordingly (L321, L373 tracked changes).

For the British Southern Ocean (BSO) cruise, only CCN concentrations inferred from nss-sulfate are available in O'Dowd et al. (1997), not comparable with any of our $N_{CCN}$.

RC2.22: Lines 300-330 - I appreciate the thorough review of previous measurements - I do find the discussion challenging to follow with the different seasons, years, SS, etc. I suggest adding a table that summarizes these aspects. Also, are all of these studies evaluating the same aerosol size range?

**AC2.22:** Reference values were summarized in Tab. 1. Generally, we only considered studies for comparison, if the cut-off ($PM_{40}$) and CCN instrumentation were comparable to our measurements. However, if not, then it is mentioned in the text (e.g., L357).

RC2.23: Figure 4 - Should the hygroscopicity parameter values be on a linear scale (as in Figure 3B)? Also, could points be added to these PDFs (and to Figure 7) to provide an idea of the bin sizes used?

**AC2.23:** The log-scale is used since the $\kappa$ formula from Petters and Kreidenweis (2007) is a function of $D_{crit}$ and the logarithm of *SS*. Fig. 4 clearly shows that our derived κ values follow a logarithmic distribution. Markers have been added to the figure to make the bins more visible. Note, there are 10 bins per order of magnitude, 30 in total.

RC2.24: Figure 5 – Are these data from the LV or HV filters? Please clarify. (same for Figures 6&7)

**AC2.24:** The captions for Fig. 5, Fig. 6, and Fig. 8 have been updated to clarify (see below).

- Fig. 5: Time series of INP number concentration ($N_\text{INP}$) at (a) –24, (b) –20, (c) –16, (d) –12, and (e) –8°C from the LV filters sampled during ACE.
- Fig. 6: Mean values (crosses) and box-and-whiskers plots indicating the median (horizontal lines), inter-quartile range (boxes), and 10th to 90th percentiles (whiskers) of INP number concentration ($N_\text{INP}$) from the LV filters sampled during Leg 1 (green), Leg 2 (orange), and Leg 3 (purple).
- Fig. 8: Normalized probability density functions (solid) and geometric mean values (dotted) for INP number concentrations ($N_\text{INP}$) at (a) –24, (b) –20, (c) –16, (d) –12 and (e) –8°C from the LV filters sampled during ACE.

RC2.25: Line 398 - Values above the upper detection limit should be excluded, see Major Comment 4.

**AC2.25:** Please see AC2.5

RC2.26: Line 403 - "assessment of air mass origin is not possible for ACE" - See Major Comment 2.

**AC2.26:** Please see AC2.3

RC2.27: Line 411 - "the derivation of N_INP was fundamentally different" - how was it different?

**AC2.27:** Unlike our method of $N_\text{INP}$ derivation, which utilizes a freezing array method, Bigg (1973) used a thermal diffusion chamber. The text has been changed accordingly (L411, L477 tracked changes).

The techniques to measure $N_\text{INP}$ were however different. Filter sampled during ACE were analysed with a freezing array method (subsection 2.3), while INP contents in Bigg (1973) were analysed by means of a thermal diffusion chamber.

RC2.28: Lines 411-415 - McCluskey et al. (2018) also discussed the hypothesis put forth by Bigg (1990) that a decadal decline in N_INP was possible over the Southern Ocean. INP concentrations reported in McCluskey et al. (2018) are also consistent with more recent measurements (McFarquar et al. 2021); It is not clear to me why so much focus is on comparing to Bigg (1973) without also considering more recent data. See Major Comment 1.

**AC2.28:** Please see AC2.2

RC2.29: Line 419 - See Major Comment 4.

**AC2.29:** Please see AC2.5

RC2.30: Line 440 - The Tobo et al. (2013) study was specific to terrestrial primary biological aerosol particles (PBAP) and I am unaware of marine INP studies that have identified a relationship between PBAP and INPs in a marine environment.

**AC2.30:** We agree that Tobo et al. (2013) focuses on particles of terrestrial origin. Studies showed that biogenic particle sampled in the Arctic act as INP at temperatures above ca. -16°C (McCluskey et al., 2018b; Hartmann et al., 2021). We assume that the same holds true for the Southern hemisphere. The text has been updated accordingly (L440, L546 tracked changes).

PBAP were found to act as INP in several studies in marine regions of the Northern hemisphere (e.g., McCluskey et al., 2018b; Hartmann et al., 2021) and we assume the same to be the case for the Southern hemisphere. Therefore, we conclude PBAP from SSA to be a potential source for the INP measured on the open ocean sections of the cruise.

References: McCluskey et al. (2018b), doi: 10.1029/2017JD028033; Hartmann et al. (2021), doi: 10.5194/acp-21-11613-2021

**RC2.31:** Line 459 - "In other words, the PDF given is representative for the marine environment" I do not think the analysis supports this statement. It is also confusing to follow this up with "Note that INP active at this temperature range can be either of mineral nature and long-range transported from terrestrial sources, or originating from marine sources."

**AC2.31:** Following RC1.5, we realised the interpretation of $N_{\text{INP},-24}$ is not helpful, as values are close to background levels indicated by the FBF. Consequently, the passage, as well as the follow-up, have been removed (L455, L565 in tracked changes).

**RC2.32:** Figure 8 - Are all samples included here, even ones with expected terrestrial influence from the ports (Grey shaded periods from Figure 5)? Suggest to segregate data using the land proximity data (i.e., data within grey shade versus not shaded from Figure 5).

**AC2.32:** Yes, samples from all parts of the cruise are included in the correlation analysis (subsection 2.5). Following the suggestion, we performed the correlation analysis with only a subset of our data set. We used a threshold in $N_{\text{INP},-16}$ ($\leq 10$ m$^{-3}$) to assess the correlation for marine background INP levels, consistent with our findings on the air-mass origin during low $N_{\text{INP},-16}$ cases (Fig. 7). This subset ($n = 75$) showed only slight differences compared to the full data set ($n = 89$) in terms of $\rho$ values. For example, between $N_{\text{INP},-12}$ and $N_{\text{INP},-20}$, a $\rho$ of 0.59 was found, which is lower than the $\rho$ of 0.73 for the full data set. Lower correlation coefficients for this subset were also found for $N_{\text{total}}$ and $N_{\text{CCN},1.0}$ (0.59 compared to 0.62) and $N_{\text{mode2}}$ and $N_{\text{CCN},0.5}$ (0.599 compared to 0.61). These 3 differences in $\rho$ are very close to our threshold for a meaningful correlation of 0.6. For the subset associated with terrestrial/coastal influence ($N_{\text{INP},-16} > 10$ m$^{-3}$), the correlation analysis could not be performed due to the low number of samples ($n = 11$) for a number of quantities. However, this does not contradict the assumption that our analysis is representative for marine background conditions.

**RC2.33:** Line 479 - Note that a similar range of INP number concentrations is reported in McFarquhar et al. (2021) that expands the seasonal and spatial representation.

**AC2.33:** Please see AC1.4

**RC2.34:** Section 3.2 - I find it unclear if the authors think INP observations are thought to be representative of "open ocean" or were influenced by terrestrial air masses, or both throughout this discussion. I think it's both (based on abstract), but overall I suggest reorganizing this to have a clear consistent message for the reader.

**AC2.34:** We find that our INP observations contain both signals from the open ocean and terrestrial/coastal sources. We hope, the inclusion of the analysis of the air-mass origin (AC2.3) helps to make things clearer.

**RC2.35:** Line 560 - What wind speeds were typical? I do not see windspeeds included anywhere. Is the relationship between sea salt or SSA and windspeed expected to be linear?

**AC2.35:** The cited literature shows that the relation between wind speed and sodium concentrations is not necessarily linear, but logarithmic. Typical values for $U_{10}$ during ACE have been added for reference. The text has been edited, to account for both (see below).

- L556 (L692 tracked changes): Other studies in the SO region found positive but non-linear connections between wind speed and sodium mass concentrations (e.g., Schmale et al., 2013; Yan et al., 2020a; Landwehr et al., 2021).
- L559 (L696 tracked changes): Note that the wind conditions encountered during ACE are characterised by median values of 9.88 (Leg 1), 6.62 (Leg 2), and 8.85 m s$^{-1}$ (Leg 3; Schmale et al., 2019) and the relative variability of the daily $U_{10}$ averages is roughly ±70 %.

RC2.36: Line 565 - Can the authors clarify how the PM10 measured size range compares to the CCN measured size range (from Line 345: "For the size range between roughly 37 and 123 nm probed with our SS (Tab S1)") and what this would mean for comparing mass concentrations from the full size range from the HV filters?

**AC2.36:** Please see AC2.10

RC2.37: Line 591 - What is meant by a "delayed connection"?

**AC2.37:** The sentence has been extended to clarify how INP abundance could respond to Chl-*a* signals in a time-shifted manner (L590, L732 tracked changes).

This suggests that the measured INP are not originating from local biogenic sources but does not exclude a time-shifted response.

RC2.38: Lines 591-598 - Are the authors aware of the existing marine INP parameterizations based on total aerosol surface area (McCluskey et al., 2018) aerosol volume (Mitts et al., 2021) - see Major Comment 3. Aerosol type-specific parameterizations may provide a lot of insight as to the INP type.

**AC2.38:** Please see AC2.4

Conclusions

RC2.39: Line 632 - "INP spectra (Fig 8) for the most part of the cruise feature similar levels and temperature range between -12 and -24degC." - What is meant by "similar levels"? I see 3 orders of magnitude variability in N_INP for any given temperature between -20 and -7degC in Figure 8. From the abstract (Lines 12-13) "[INP] concentrations spanned up to 3 order of magnitude, e.g., at -16degC from 0.2 to 100 m-3.".

**AC2.39:** We refer to the spectra showing steadily increasing $N_{INP}$ with decreasing temperature for almost every filter. This is in contrast to the feature of strong increases $N_{INP}$ within a small temperature range mentioned later (see below).

- L588 tracked changes: A typical, steady increase in $N_{INP}$ with decreasing temperature (1 order of magnitude per 5K) can be observed for the majority of filters and the FBF's curve (pink line).
- L632 (L785 tracked changes): INP spectra (Fig. S8a) for the most part of the cruise feature a steady increase in INP concentration with decreasing temperature.

**RC2.40:** Lines 634 - "We interpret this signal [correlations between N_INP in the temperature range between -12 and -24degC] as indication that mixed long-range-transported populations of INPs of biogenic origin (T>-20 deg C) and mineral dust (T <. -20degC) were present. " - Again, it is not clear to me that this statement is supported by evidence available in this study.

**AC2.40:** As evidence for the sampling of biological INP, we can only point to several filters showing features in the spectra of increased concentrations at high temperatures ("bio bumps"). Since there was no thermal-treatment-check performed with the filters, there is no proof for the sampling of biological particles acting as INP. The following text has been added to emphasise on the hints we see regarding the sampling of particles of biological origin acting as INP (see below).

- L589 tracked changes: At temperatures above −20°C, a feature of sudden increase in curve steepness for a number of filters is apparent. This feature is unlike the previously mentioned, steady increase in $N_{INP}$. A high concentration of INP above −15°C is typically associated with a signal of a biological INP source mixed with a mineral or less efficient INP source (e.g., Creamean et al., 2019).
- L786 tracked changes: Features in the spectra of increased $N_{INP}$ at warm temperatures indicate warm-temperature INP which are connected to biological origin (e.g., Creamean et al., 2019).

References: Creamean et al. (2019), doi: 10.5194/acp-19-8123-2019

---

## Author Response (AR2)

RC = Reviewer Comment, AC = Author Comment, new suggested text in blue

Note, line numbers refer to the original manuscript, the location in the revised can be seen best in the tracked changes version document. We indicate these line numbers in ().

**Answers to Reviewer 1**

Anonymous referee #1, 01 Apr 2022

Summary

RC1.1: The authors have satisfactorily addressed the main comments from my initial review. The manuscript now includes a better introduction with reference to the studies of Schmale et al. (2019) and Welti et al. (2020), the comparison with McCluskey et al. (2018) has received more attention, and the discussion of the PDFs has also been greatly improved by linking them to trajectory results shown in Fig. 7. I still found a number of mostly technical corrections, the list below may not be complete, and I would encourage further careful proofreading.

**AC1.1:** We thank the reviewer for the positive review. The proposed technical corrections have been addressed.

RC1.2: P2L43/44: "… it is called a cloud condensation nucleus."

**AC1.2:** The sentence has been edited as suggested (L44, L44 tracked changes).

Particles acting as nuclei for cloud droplet formation at atmospherically relevant (water vapour) supersaturation are commonly referred to as cloud condensation nuclei (CCN).

RC1.3: P3L85: The sentence starting with "Two recent …" is not complete.

**AC1.3:** The sentence has been changed in order to follow a complete grammatical structure (L86, L87 tracked changes).

Two recent cruises were conducted as part of the Cloud, Aerosols, Precipitation, Radiation and Atmospheric Composition campaign (CAPRICORN-I & II).

RC1.4: P3L87: It should be "varied" instead of "varying"

**AC1.4:** The sentence has been edited as suggested (L87, L88 tracked changes).

For CAPRICORN-I, observed $N_{INP}$ over the SO in the temperature range between -12 and -31°C varied between 0.04 and 1000 m$^{-3}$ (McCluskey et al., 2018a).

RC1.5: P9L246: Maybe: "Integration gives the total particle surface area and volume for each measurement."

**AC1.5:** This part, including the suggested sentence, has been updated (L245, L246 tracked changes).

For this, the particle surface area and volume size distributions were first inferred from the number size distribution, assuming spherical particles, and then integrated over the entire diameter range. This was done for each size distribution measurement.

**RC1.6:** P9L248: I am bit wondering about the variable name "v_s". I see that it is also used in Mitts et al. (2021), but wouldn't the names "n_s" for the surface density of active sites, and "n_v" for the volume density of active sites be more appropriate? The same applies to the labelling of the y-axes in Figs. 8b and c (which again is the same as in Mitts et al.), why is "n_s" referred to as number site density and "v_s" as volume site density? I would call them "surface density of active sites" or "number of active sites per unit surface area of INPs" respectively "volume density of active sites" or "number of active sites per unit INP volume". But please correct me if I have a misunderstanding here.

**AC1.6:** We agree that $n_v$ (instead of $v_s$) as the symbol for the volume site density is a better choice. The text and Fig. S8 have been changed accordingly (see below).

- L247 (L250 tracked changes): These values were averaged over the 8 h sampling time of each LV filter and $N_{INP}$ is divided by these values, resulting in the ice active site density ($n_s$) and ice active volume density ($n_v$), respectively.
- L548 (L556 tracked changes): The resulting spectra of ice-active number site density, $n_s$, and volume site density, $n_v$, are given in Fig. S8b,c.
- L556 (L563 tracked changes): The range of $n_v$ reported in Mitts et al. (2021) are included in Fig. S8c for comparison.
- L10 in the SI: Figure S8 shows the temperature-dependence of $N_{INP}$, $n_s$, and $n_v$ for the LV filters.
- Fig. S8 caption (P12 in the SI): Temperature-dependence of (a) INP number concentration $N_{INP}$, (b) number site density $n_s$, and (c) volume site density $n_v$ for the LV filters sampled during ACE. Values of $n_s$ ($n_v$) were calculated by normalising $N_{INP}$ with the total particle surface area (total particle volume) derived from an averaged particle number size distribution per filter under the assumption of a population of only spherical particles.

**RC1.7:** P13L351: Incomplete sentence. Maybe: "The averaging of the CGBS measurements … gives a median of …" And delete "was found" at the end.

**AC1.7:** The sentence has been revised to make it complete (L353, L356 tracked changes).

The averaging of the CGBS measurements which feature only baseline conditions gives a median of ~130 cm⁻³ (triangle pointing right in Fig. 3a).

**RC1.8:** P14L361: "are reported in Sanchez et al."

**AC1.8:** The sentence has been edited as suggested (L362, L365 tracked changes).

As for aircraft-based CCN measurements, $N_{CCN,0.3}$ between 17 and 264 cm⁻³, with an average of 123 ± 58 cm⁻³ (mean ± SD), are reported in Sanchez et al. (2021) for flights through the MBL between 42.5–62.1° S and 133.8–163.1° E during the Southern Ocean Clouds, Radiation, Aerosol Transport Experimental Study (SOCRATES).

**RC1.9:** P14, header of Table 1: Duplicate header label "Time frame"

**AC1.9:** Tab. 1 has been changed accordingly.

**RC1.10:** P17L431: Sentence is also difficult to read, maybe: "INP concentrations outside the detectable range (indicated by triangles and estimated as described in subsection 2.3) were included

in the figure as they mark …" Why do the maximum INP concentrations represent the lower detection limit and vice versa?

**AC1.10:** We agree that this sentence needs to be improved in terms of readability. As for the representation of the detectable range of $N_{INP}$, the text was misleading. A value outside the detectable range is represented by the maximum (minimum) value of the lower (higher), unresolvable concentration range. The sentence has been edited as suggested and the idea behind giving the detection limit has been clarified (L433, L437 tracked changes).

INP concentrations outside the detectable range (indicated by triangles and estimated as described in subsection 2.3) are represented in the figure by each filter's lowest (lower detection limit) and highest resolvable concentration value (upper detection limit).

RC1.11: P19L462: Why does the exception also apply to -20°C? For -16°C it is clear to me (Fig. 5c), as your measurements on the open ocean nicely fall into the range of the McCluskey data, but for -20°C (panel b) it seems to me that they are also at the upper end of the CAPRICORN-I data.

**AC1.11:** We agree that this is not the case at -20°C. The sentence has been edited accordingly (L464, L469 tracked changes).

At each temperature, $N_{INP}$ observed during ACE are at the upper end or higher than concentrations observed during CAPRICORN-I, except at -16°C when low concentrations were measured on the open ocean in air-masses without terrestrial influence.

RC1.12: P19L470/471: Maybe: "have not been identified so far"

**AC1.12:** The sentence has been edited as suggested (L473, L478 tracked changes).

However, potential mechanisms behind such a hypothetical feedback have not been identified so far.

RC1.13: P22L535/536: The description "a feature of sudden increase in curve steepness" was not particularly clear to me, perhaps it would be better to speak of a shoulder in the INP spectrum at higher temperatures, which could indicate biological INP sources.

**AC1.13:** We agree that "shoulder" is a simpler description of the feature commonly known as "bio-bump". This part of the text has been updated accordingly (L539, L545 tracked changes).

At higher temperatures (>-20°C) a shoulder in the INP spectra of a number of filters is apparent.

RC1.14: P23L544/545: Please explain more clearly what you mean by "which can include different approaches to INP number derivation, independent of N_INP". As I understand it, n_s would just be a concept to normalize the results of studies where different particle sizes were used.

**AC1.14:** We agree that the text was not clear enough on why the normalisation is used. The sentence has been edited as suggested (L547, L554 tracked changes).

Normalisation enables comparison of INP properties across different studies, which can include different $N_{INP}$ derivation approaches or sampled particle size ranges, independent of $N_{INP}$.

RC1.15: P23L558: Better phrase it like e.g.: "For the sake of completeness, however, we also show

in Fig. S5 the additional INP spectra determined from the HV samples (…).”

**AC1.15:** The sentence has been edited as suggested (L561, L569 tracked changes).

For the sake of completeness we give in Fig. S5 the additional INP spectra determined from HV samples (*DHA-80* sampler, see subsection 2.3).

RC1.16: P26L641: Repetitive "The lack of a correlation". Maybe just state: "This might be a smearing effect …"

**AC1.16:** The sentence has been edited as suggested (L646, L654 tracked changes).

This might be a smearing effect due to averaging, since the highest $N_{\mathrm{CCN}}$ (subsection 3.1) coincide with the highest $M_{\mathrm{MSA}}$ (subsection 3.3) when the cruise was in the vicinity of the Antarctic coast (Leg 2).

RC1.17: P27L652: You added "that" after "supports" in the revised version, but I don't think this is correct.

**AC1.17:** The sentence has been edited as suggested (L657, L665 tracked changes).

This again supports the observation of SSA particles not significantly contributing to the CCN population, as SSA dominates the sampled particle mass but not the particle number.

RC1.18: P29L713: "The comparison with other studies shows that the …"

**AC1.18:** The sentence has been edited as suggested (L718, L726 tracked changes).

The comparison with other studies shows that the concentrations observed during ACE are lower than what was observed previously over the SO (Bigg, 1973), while being on the upper end of what is reported in McCluskey et al. (2018a) for a specific sector of the SO in March–April 2016 (Fig. 5 and Fig. S8a).

**Answers to Reviewer 2**

Anonymous referee #2, 06 Apr 2022

Summary

RC2.1: In the paper submitted titled "Circum-Antarctic abundance and properties of CCN and INP", Tatzelt et al. describe results from the 2016/2017 research voyage that aimed to characterize southern ocean CCN and INPs. The authors have addressed several comments, including an impressive back trajectory analysis that really add to the discussion. Below are several minor comments and one major comment that I will provide related to the inclusion of background INP number concentrations.

**AC2.1:** We thank the reviewer for the positive and constructive review. The inclusion of the field blank filter INP number concentrations was carefully taken into consideration. We have made changes to the text following the major and general comments below.

Major comment:

RC2.2: Lines 237-239: "Based on the fice of the field blank filters (FBF) we determined averaged temperature-dependent INP concentrations, NINP;FBF, which are given as point of reference for background concentration levels whenever NINP for the sampled filters are shown." -yet, the FBF results are only shown in the supplemental figures (Figures S5 and S8) and are not included in figures 5, 6, or 8. I do not think adding the FBF to these figures will significantly change the results of the paper. However, not including these FBF results carefully could lead to mis-use of the data if someone were to use these data in future comparisons to other observations or model estimates in the future. Given that many of the measured values reported are below the background levels measured from the FBF, I think this is important to address and suspect it will also be straight-forward to add.

**AC2.2:** We agree that the addition of the FBF-derived $N_{INP}$ in the INP figures gives context to the presented, encountered values. Consequently, the mean $N_{INP,FBF}$ has been added to Fig. 5 and Fig. 6 as suggested and the respective captions have been updated (see below). As for Fig. 8 (PDF of $N_{INP}$), we decided against the inclusion of the mean $N_{INP,FBF}$ due to a poor readability of the resulting figure. We hope, the message that our values are at times close to the background comes across in the other two figures regardless.

- Fig. 5 caption (P18): In (a–d) the measurement background from averaged spectra of field blank filters (FBF) is indicated (dash-dotted lines).
- L435 (L438 tracked changes): For $T \leq$ -12°C (Fig. 5a—d), the respective measurement background INP concentrations are represented via the averaged FBF spectra (dash-dotted lines), as described in subsection 2.3.
- Fig. 6 caption (P20): In the figure, the measurement background is represented by the averaged spectra of the field blank filters (FBF; dash-dotted lines) and the number of data points ($n$) are indicated.
- L484 (L491 tracked changes): As a point of reference for the measurement background, concentrations of the averaged FBF are included (dash-dotted lines).

Typos/minor comments:

RC2.3: L84 – It might be helpful to include additional information on the freezing mode of focus of this work.

**AC2.3:** We agree that immersion freezing, the freezing process used to investigate the sampled INP, needs direct statement in the text. Information on the used freezing method has been added the abstract (L11, L11 tracked changes).

Ambient INP number concentrations were measured in the temperature range from -5 to -27°C using an immersion freezing method.

RC2.4: L85-86 – "two recent cruises:…" is not a complete sentence.

**AC2.4:** Please see AC1.3

RC2.5: L94 – "… whether sampled INP have terrestrial or oceanic sources.." I think the word "have" should be replaced with the word "had"

**AC2.5:** The sentence has been edited as suggested (L95, L96 tracked changes).

Correlation of INP and ambient radon concentration was used to assess whether sampled INP had terrestrial or oceanic sources for the CAPRICORN-I cruise.

RC2.6: L134 – ".. wind direction within a 180deg half-circle at the sampler, with the RV's exhaust at 90deg" – I find this very challenging to follow. I think you mean that the exhaust was behind the sampler and the 180deg half-circle is in front the sampler? Can you please clarify?

**AC2.6:** We agree that description of the exclusion wind directions was not straightforward. The sentence has been edited to improve understandability (L135, L136 tracked changes).

Sampling was stopped automatically during periods with wind direction within a half-circle at the sampler centred towards the stack exhaust situated on the RV's stern.

RC2.7: L240/Table S3 – why are the HV FBF results not included in Table S3 for comparison?

**AC2.7:** We agree that the addition of the FBF INP concentrations for the HV samples will give context to the values presented, e.g. in Tab. S4. These values have been added to Tab. S3 and the respective caption has been updated accordingly.

- L241 (L242 tracked changes): In Tab. S3 $N_{\text{INP,FBF}}$ is given for the LV and the HV filters.
- Tab. S3 caption (P14 in the SI): Mean INP number concentration of field blank filters (FBF) for LV sampling ($N_{\text{INP,LV,FBF}}$) and HV sampling ($N_{\text{INP,HV,FBF}}$) at selected temperatures ($T$).

RC2.8: L334 – "… considered representative for the whole SO region" – Important to mention that this is specific to the SO marine boundary layer and summertime. This statement might also fit better after the discussion of the previous CCN measurements?

**AC2.8:** We agree that the statement made in this sentence was too general. However, since the conclusions are drawn directly from what is shown in Fig. 3a, we think the sentence would not fit at an earlier point in the text. The sentence has been edited to include specifications on height and season (L335, L338 tracked changes).

With this, the CCN concentrations given in Tab. S1 can be considered representative for the MBL over the whole SO region during summertime.

RC2.9: L346 – "For this period, a median of ~230cm-3" – I think this referring to N_CCN,0.5, right? Can you please specify?

**AC2.9:** We agree that there is specification needed at this point. The sentence has been edited to indicate the $SS$ of the $N_{\text{CCN}}$ in question (L348, L351 tracked changes).

For this period, a $N_{\text{CCN,0.5}}$ median of ~230 cm$^{-3}$ is reported (triangle pointing left in Fig. 3a).

RC2.10: Table 1 – note that the second column header (Time frame) has a typo in that it is redundant. Also, can the authors add if these studies were all summertime measurement campaigns in the caption?

**AC2.10:** The duplicate in the header of the second column has been removed (see AC1.9). Information on the common season of the presented studies has been added to the caption of Tab. 1 as suggested (P14, P14 tracked changes).

Overview of a selection of studies on aerosol particles and CCN over the summertime Southern Ocean.

RC2.11: L448 – "Contrary to these regions, air-masses passing over Antarctica did not show higher NINP than oceanic air-masses (Fig. S7)." – I recognize the authors are referring to Figure S7, but what about the period of elevated Antarctic airmass influence and elevated INP number concentrations between the January 26 and Feb 9 timestamps in Figure 5?

AC2.11: Looking into the data for Fig. S6d, we found elevated Antarctic air-mass influence ($\geq$80%) for the period 28$^{th}$ January to 31$^{st}$ January 2017. This coincides with the RV being close to or stationed at Mertz Glacier (see cruise report in Walton & Thomas, 2018). The mentioned elevated INP concentrations (e.g. $N_{INP,-16}$>10 m$^{-3}$ in Fig. 5c) are found for a sample from 2$^{nd}$ February 2017, when the RV was already on the Ross Sea. Further, Fig. S6c shows for 2$^{nd}$ February 2017 a strong influence of sea ice and the MIZ on the air-mass origin.

References: Walton & Thomas (2018), doi: 10.5281/zenodo.1443511

RC2.12: L453 – "Filter sampled during ACE …" should this be filters (plural?)

AC2.12: The sentence has been edited as suggested (L456, L461 tracked changes).

Filters sampled during ACE were analysed with a freezing array method (subsection 2.3), while INP contents in Bigg (1973) were analysed by means of a thermal diffusion chamber.

RC2.13: L478 – "we report all averaged values…" – where are the average values for option A? Is that in Table S2?

AC2.13: We agree that the text was not clear enough on which approach led to $N_{INP}$ (option A) and $N_{INP}^*$ (option B) in Tab. S2. This part of the text and the caption of Tab. S2 have been updated to be more precise (see below).

- L476 (L482 tracked changes): Two different approaches were used for averaging the INP concentrations of the LV samples. In the first approach only values which are inside the detectable range are considered and the averages are given as $N_{INP,LV}$ in Tab. S2. For the second approach, values outside the detectable range ($N_{INP,LV}^*$) were included by using a value on the edge of the detectable range instead (see subsection 2.3). Results of the two approaches differ in mean, median, and geometric mean concentration values by up to ±50 %. The largest differences were found at a $T$ of -8 and -24°C, where the number of data points outside the detectable range is largest.
- Tab. S2 caption (P13 in the SI): For this averaging only concentrations within the detectable range are considered and the number of samples, $n$, are indicated in the table.

RC2.14: L497 – ".. FBF concentrations are 0.08…" why were only the FBF concentrations mentioned for -12 and -16? The values for -20 and -24 deg C are 7.72 m-3 and 57.76 m-3 (Table S3), respectively, which falls very near the peak of the PDF in panels a and b. Please see major comment 1.

AC2.14: Please see AC2.2

RC2.15: L559-571 – somewhere in this discussion the authors should also discuss the higher background concentrations of the HV filters (Figure S5) compared to the LV filters (Table S3 and Figure S8). From eye, the FBF results associated with the HV filters are an order of magnitude

(factor of 2) higher than the LV filters for temperatures –12 and -16 deg C (-20 and -24 deg C), which would limit the ability to measure lower INP number concentrations, thereby making the spread more narrow and biased high. I think the existing discussion is also helpful, but this large difference in field blank background INP concentrations is likely important.

**AC2.15:** We agree that from the higher measurement background for the HV sampling one can assume the measured concentrations to be naturally higher. We included this idea in the text (L571, L580 tracked changes).

The higher background INP levels for the HV filters, indicated by higher $N_{\text{INP,FBF}}$ compared to the LV sampling (Tab. S3), hints on a limited ability of the HV sampling to measure lower INP number concentrations and in consequence overall higher measured INP number concentrations.

RC2.16: L581 – I think this should be Table S5.

**AC2.16:** The sentence has been edited as suggested (L585, L593 tracked changes).

Averaging sodium mass concentrations for the whole cruise gives a median value of 2.8 µg m⁻³, with an IQR from 1.8 to 3.9 µg m⁻³ (Tab. S5).

RC2.17: Figure 7 caption: I think there is a typo, where the non-Antarctic land masses should be red, rather than light blue.

**AC2.17:** The caption of Fig. 7 has been edited as suggested (P21, P21 tracked changes).

Color codes for surface types are: non-Antarctic land masses (red), non-Antarctic coastal regions or islands (orange), Antarctic continent or coastal regions (ANT; yellow), ice-covered regions (light blue), and open ocean (dark blue).

Supplemental minor comments:

RC2.18: L15 – should the Thurnherr referenced be mentioned here?

**AC2.18:** The sentence has been edited as suggested (L15 in the SI).

Backward-trajectories for ACE are available in Thurnherr et al. (2020).

Reference: Thurnherr et al. (2020), doi: 10.5281/zenodo.4031705

RC2.19: L28 – "Here, the model's land-sea mask" – which model?

**AC2.19:** The LAGRANTO tool uses ERA-interim reanalysis data which is the output of the ECMWF Integrated Forcasting System model. The text has been updated to include this information (see below).

- L15 in the SI: Backward-trajectories for ACE are available in Thurnherr et al. (2020). They used the Lagrangian analysis tool (LAGRATO) described in Sprenger and Wernli (2015) with reanalysis data from the European Centre for Medium Range Weather Forcasts (ECMWF). These reanalyses are produced by the ECMWF Integrated Forecasting System (IFS), an atmospheric model and data assimilation system.
- L31 in the SI: Here, the IFS's land-sea mask is used for initial classification of the surface signal type.

References: Sprenger and Wernli (2015), doi: 10.5194/gmd-8-2569-2015

**RC2.20:** L33 – "A commonly use sea ice fraction…" – is there a reference for this?

**AC2.20:** In our understanding, the commonly used approach of classifying grid boxes in atmospheric models as open ocean if the sea ice fraction of the grid box is below 15 % originates from Cavalieri et al. (1991) finding the best agreement for the sea ice edge location between airborne microwave radiometer observations and ship-based crossings for a sea ice concentration of 15%. The reference has been added to clarify the origin of the threshold value (L36 in the SI).

A commonly used sea ice fraction threshold of <15 % is applied here to classify as "open ocean" (following Cavalieri et al., 1991).

References: Cavalieri et al. (1991), doi: 10.1029/91JC02335

**RC2.21:** Figure S6 – Panel a is confusing to me. It is unclear how 10 hour trajectories would relate to "duration of trajectory cluster" values larger than 240 hours, and here the values go up to 1920 hours? I think I may be missing something.

**AC2.21:** For the airmass-origin analysis of the LV samples, the hourly clustering (and subsequent averaging) of all trajectories ending within the PBL was performed up to eight times, corresponding to the 8 h sampling interval of the LV filters. The maximum duration of 1920 h is reached in the case that each of the eight hourly time steps consists of a trajectory cluster with the maximum duration of 10 d (or 240 h). The text has been updated in an attempt to improve the explanation of the clustering procedure (see below).

- L19 in the SI: At an hourly resolution, trajectories are available for 56 pressure levels (between the surface and 500 hPa above sea level) above the RV's position. To achieve more robust statistics, for each hour all trajectories ending within the planetary boundary layer (PBL) above the RV's position are averaged into a cluster by calculating median values of latitude, longitude, pressure level, boundary layer pressure level, total precipitation, land fraction and sea ice fraction.
- L41 in the SI: The final step of the analysis is the averaging of PBL signals from all clusters within each LV filter's 8 h sampling period. The total number of trajectories that are averaged in clusters and considered per LV filter is between 1 and 28, with a mean of roughly 15.

**RC2.22:** Table S2 – I think additional details are needed on the first grouping of averages.

**AC2.22:** Please see AC2.13

**RC2.23:** Table S3 – As mentioned above, please add the HV FBF results. Also, should there be a range since more than 1 FBF was collected (L197 of main text) and a range of sampled volume was estimated for this calculation (L239-240)?

**AC2.23:** The $N_{\text{INP,FBF}}$ for the HV samples have been added to Tab. S3 (see AC2.7). We found the $N_{\text{INP}}$ of individual FBF to vary between the range described by a factor 2 of the mean FBF. Given that the variability of $V_{\text{flow}}$ is roughly 10%, we expect resulting variability for $N_{\text{INP,FBF}}$ in the same region. This is well within our conservative estimation for the uncertainty in FBF of a factor 2. With that we think it sufficient to show FBF ± 2 FBF in the spectra plots of Fig. S5 and Fig. S8.